# GROUND-A-VIDEO: ZERO-SHOT GROUNDED VIDEO EDITING USING TEXT-TO-IMAGE DIFFUSION MODELS

**Hyeonho Jeong & Jong Chul Ye**
Kim Jaechul Graduate School of AI, KAIST
{hyeonho.jeong,jong.ye}@kaist.ac.kr

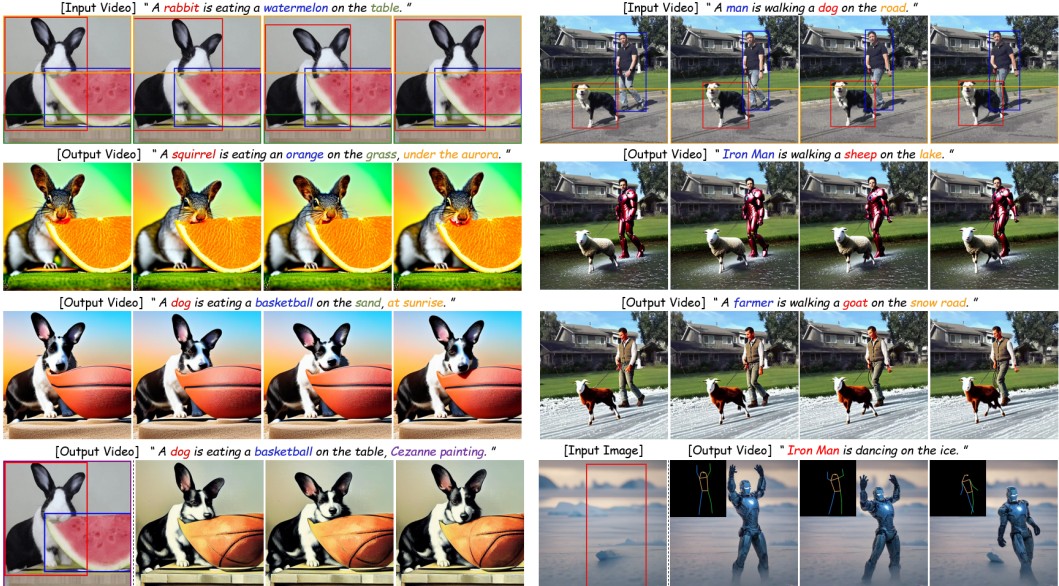

Figure 1: Ground-A-Video achieves multi-attribute editing, video style transfer with attribute change, and text-to-video generation with pose guidance, all in a time-consistent and training-free fashion. The boxes in the right-bottom images visualize the series of pose guidance.

## ABSTRACT

Recent endeavors in video editing have showcased promising results in single-attribute editing or style transfer tasks, either by training text-to-video (T2V) models on text-video data or adopting training-free methods. However, when confronted with the complexities of multi-attribute editing scenarios, they exhibit shortcomings such as omitting or overlooking intended attribute changes, modifying the wrong elements of the input video, and failing to preserve regions of the input video that should remain intact. To address this, here we present a novel grounding-guided video-to-video translation framework called Ground-A-Video for multi-attribute video editing. Ground-A-Video attains temporally consistent multi-attribute editing of input videos in a training-free manner without aforementioned shortcomings. Central to our method is the introduction of Cross-Frame Gated Attention which incorporates groundings information into the latent representations in a temporally consistent fashion, along with Modulated Cross-Attention and optical flow guided inverted latents smoothing. Extensive experiments and applications demonstrate that Ground-A-Video's zero-shot capacity outperforms other baseline methods in terms of edit-accuracy and frame consistency. Further results and code are available at *http://ground-a-video.github.io*.

# 1 INTRODUCTION

Coupled with massive text-image datasets (Schuhmann et al., 2022), diffusion models (Sohl-Dickstein et al., 2015; Ho et al., 2020; Song et al., 2020b) have revolutionized text-to-image (T2I)

generation, making it increasingly accessible to generate high-quality images from text descriptions. Additionally, the domain has seen profound expansion into several subfields, including controlled generation and real-world image editing. On the other hand, the endeavor to extend the success to the video domain poses a significant computational hurdle. Attaining time-consistent and high-quality results necessitates training on expensive video datasets—an endeavor beyond the means of most researchers, particularly given the absence of publicly available, generic text-to-video models.

As such, pioneering approaches exhibit promise in text-to-video generation (Ho et al., 2022b;a) and video editing (Esser et al., 2023) by repurposing T2I diffusion model weights for extensive video data training. Specifically, in pursuit of cost-effective video generation, Wu et al. (2022) suggests fine-tuning the T2I model on a single video, which enables generating variations of the video. Similar to the practice of manipulating attention maps within the realm of image editing (Hertz et al., 2022; Tumanyan et al., 2023; Parmar et al., 2023), various methods guide the denoising process by self-attention maps (Ceylan et al., 2023), cross-attention maps (Liu et al., 2023; Wang et al., 2023), or both (Qi et al., 2023), which are obtained during the input video inversion stage. Recently, to incorporate the denoising process with additional structural cues, ControlNet (Zhang & Agrawala, 2023) has been transferred to video domain, achieving structure-consistent output frames in video generation (Khachatryan et al., 2023a) and translation (Hu & Xu, 2023; Chu et al., 2023; Chen et al., 2023; Zhang et al., 2023).

Nonetheless, in the scenario of fine-grained video editing involving multiple attribute changes, i.e. $\Delta\tau = \{\tau_a \rightarrow \tau_{a'}, \tau_b \rightarrow \tau_{b'}, \tau_c \rightarrow \tau_c, \ldots, \varnothing \rightarrow \tau_{new}\}$, where $\tau$ represents a specific attributed indexed by subscript, they encounter issues of degraded frame consistency, severe semantic misalignment (Park et al., 2023), or both, as depicted in Fig. 2-*Left* and Sec. E. In specific, instances of neglecting intended attribute edits ($\tau_a \rightarrow \tau_a$), modifying the wrong elements ($\tau_a \rightarrow \tau_{b'}$), mixing two separate edits ($\tau_a \rightarrow \tau_{a'} \cdot \tau_{b'}$), and struggling to preserve regions that should remain unchanged ($\tau_c \rightarrow \tau_{c'}$) are observed. This is because in the existing models, the Cross-Attention layer is the single domain where the complex semantic changes wield their influence, where the list of intricate changes being entangled as a form of "one-sentence target prompt" makes the problem worse.

The key to address this issue lies in spatially-disentangled layout information, comprising bounding box coordinates and textual captions, namely 'groundings'. Groundings disentangle the complex semantic combination by localizing each semantic element with precise location. Recently, grounding has been successfully employed to text-to-image generation tasks. Li et al. (2023b) and Yang et al. (2023) finetune existing T2I models to adhere to grounding conditions using box-image paired datasets, while Xie et al. (2023) achieves training-free box-constrained image generation by injecting binary spatial masks into the cross-attention space. However, unlike the literature on single-image synthesis, guiding video generation process with groundings alone could significantly detriment frame consistency. One major problem is that localizing the bounding box is insufficient for the smooth frame transition (Fig. 2-*right*), which calls the need for additional structural guidance. Consequently, we propose to integrate two distinct modalities: 1) spatially-continuous conditions, including depth and optical flow maps, are employed to maintain consistent structure across frames; 2) spatially-discrete conditions, specifically 'groundings', enable precise localization and editing of each attribute within the source video. The principal contributions of `Ground-A-Video` are summarized as follows:

- To our knowledge, we present the first groundings-driven video editing framework, also marking the first instance of integrating both spatially-continuous and discrete conditions.
- We propose a novel Modulated Cross-Attention mechanism which efficiently enables interactions between differently optimized unconditional embeddings.
- To further enhance consistency, we suggest smoothing inverted latent representations using optical flow information, which can be employed following any type of inversion.
- Extensive experiments and applications demonstrate the effectiveness of our method in achieving time-consistent and precise multi-attribute editing of input videos.

## 2 BACKGROUND

**Stable Diffusion.** Distinct from traditional diffusion models (Ho et al., 2020), Stable Diffusion (SD) functions within a low-dimensional latent space, which is accessed via VAE autoencoder $\mathcal{E}, \mathcal{D}$ (Kingma & Welling, 2013). More precisely, once the latent representation $z_0$ is obtained by com-

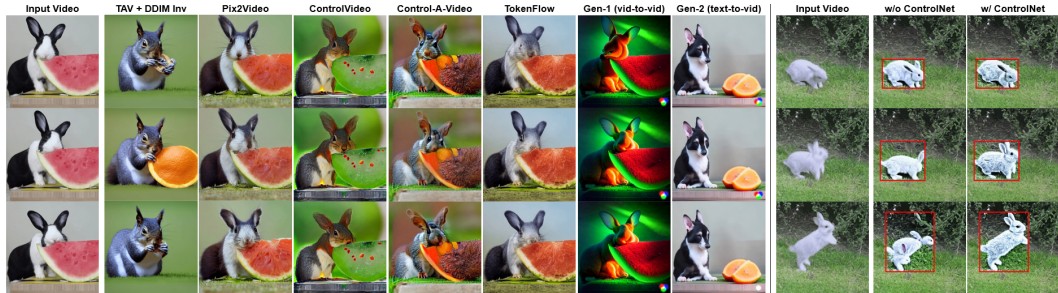

Figure 2: *Left*: Failure cases of multi-attribute video editing by various methods, driven by the target text "*A squirrel is eating an orange on the grass, under the aurora.*" Ground-A-Video's successful result for the same task is shown in the second row of Fig. 1. *Right*: Input video reconstruction with and without ControlNet.

pressing an input image $f \in \mathbb{R}^{H \times W \times 3}$ through the encoder $\mathcal{E}$, i.e. $z_0 = \mathcal{E}(f)$, diffusion forward process gradually adds Gaussian noise to $z_0$ to obtain $z_t$ through Markov transition with the transition probability:

$$q(z_t|z_{t-1}) = \mathcal{N}(z_t; \sqrt{1 - \beta_t} z_{t-1}, \beta_t I), \quad t = 1, \ldots, T, \tag{1}$$

where the noise schedule $\{\beta_t\}_{t=1}^T$ is an increasing sequence of $t$ and $T$ is the number of diffusion timesteps. Then, the backward denoising process is given by the transition probability:

$$p_\theta(z_{t-1}|z_t) = \mathcal{N}(z_{t-1}; \mu_\theta(z_t, t), \sigma_t^2 I), \quad t = T, \ldots, 1. \tag{2}$$

Here, the mean $\mu_\theta(z_t, t)$ can be represented using the noise predictor $\epsilon_\theta$ which is learned by the minimization of the MSE loss with respect to $\theta$: $\mathbb{E}_{f,\tau,\epsilon \sim \mathcal{N}(0,I),t} \|\epsilon - \epsilon_\theta(z_t, t, \tau)\|_2^2$, where $\epsilon$ refers to the zero mean unit variance Gaussian noise vector, and $\tau = \psi(\mathcal{T})$ is the embedding of a text $\mathcal{T}$.

Specifically, a prevalent approach in diffusion-based image editing is to use the deterministic DDIM scheme (Song et al., 2020a) to accelerate the sampling process. Within this scheme, the noisy latent $z_T$ can be transformed into a fully denoised latent $z_0$:

$$z_{t-1} = \sqrt{\frac{\alpha_{t-1}}{\alpha_t}} z_t + \left( \sqrt{\frac{1 - \alpha_{t-1}}{\alpha_{t-1}}} - \sqrt{\frac{1 - \alpha_t}{\alpha_t}} \right) \epsilon_\theta, \quad t = T, \ldots, 1, \tag{3}$$

where $\alpha_t$ is a reparameterized noise scheduler.

**Null-text Optimization.** To augment the effect of text conditioning, Ho & Salimans (2022) have presented the classifier-free guidance technique (cfg), where the noise prediction by $\epsilon_\theta$ is also carried out unconditionally, namely by 'null text'. Then the null-conditioned prediction is extrapolated with the text-conditioned prediction to produce the classifier-free guidance prediction:

$$\tilde{\epsilon}_\theta(z_t, t, \tau, \varnothing) = w \cdot \underbrace{\epsilon_\theta(z_t, t, \tau)}_{\text{conditional prediction}} + (1 - w) \cdot \underbrace{\epsilon_\theta(z_t, t, \varnothing)}_{\text{unconditional prediction}}, \tag{4}$$

where $\varnothing = \psi(\text{" "})$ denotes the embedding of a null text and $w$ denotes the guidance scale. However, when coupled with cfg featuring large guidance scale $w \gg 1$, DDIM inversion accumulates errors during the denoising process, resulting in flawed image reconstruction. In order to fix these errors, Mokady et al. (2023) optimizes the embeddings of a null text. First, DDIM inversion trajectory $\{z_t^*\}_{t=0}^T$ and the predicated backward trajectory $\{\bar{z}_t\}_{t=0}^T$ are computed. Then, at each timestep starting from $T$, unconditional embeddings $\{\varnothing_t\}_{t=1}^T$ are tuned towards minimizing $\|\bar{z}_{t-1} - z_{t-1}^*\|_2^2$.

## 3 METHOD

### 3.1 GROUND-A-VIDEO: OVERVIEW

Given a series of source video frames $f^{1:N}$ and a list of intended edits $\Delta\tau$, our goal is to accurately edit various attributes of the source video while avoiding unwanted modifications, in a zero-shot yet time-consistent manner. Fig. 3 illustrates the overall architecture of Gound-A-Video to achieve

this. Initially, we automatically acquire grounding information through GLIP (Li et al., 2022). The groundings and the source prompt are manually refined to form target groundings and target prompt, commonly following $\Delta\tau$. On the other branch, the input frames undergo individual DDIM inversion and null optimization, followed by optical flow-based smoothing to form smoothed latent features. The smoothed latents are separately fed into the inflated SD and ControlNet, which are modified with a sequence of attentions to achieve the temporal consistency of the video editing. The inflated ControlNet takes an additional input of depth maps. Subsequently, the target groundings are directed to the Cross-Frame Gated Attentions of the SD model, while the target prompt and optimized null-embeddings are channeled into the Modulated Cross Attentions of both SD and ControlNet models.

In Sec. 3.2, we discuss our inflated SD with per-frame inversion approach and proposed attention mechanisms (Modulated Cross-Attention & Cross-Frame Gated Attention). In Sec. 3.2, we introduce the inflated ControlNet to incorporate a spatially-continuous prior of input video: a sequence of depth maps. Lastly in Sec. 3.4, we present optical flow guided smoothing which is applied to the inverted latent features before they are fed into the SD and ControlNet models, to further improve the consistency across frames.

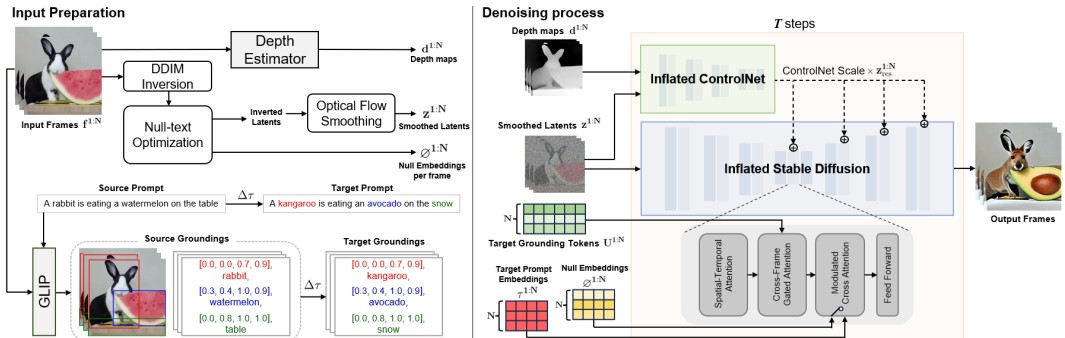

Figure 3: *Left*: **Input preparation**. We automatically obtain video groundings of input video frames $f^{1:N}$ via GLIP. This is followed by a handcraft editing phase for both the groundings and the source prompt. The input frames undergo individual inversion and null optimization, followed by optical flow-based smoothing. Furthermore, ControlNet input $d^{1:N}$ are obtained via ZoeDepth estimator. *Right*: **Denoising process**. The smoothed latents $z^{1:N}$ are fed into the inflated SD and ControlNet. The target grounding tokens $U^{1:N}$ are directed to Cross-Frame Gated Attention, while context vectors $\{\tau^{1:N}, \varnothing^{1:N}\}$ are directed to Modulated Cross Attention. The series of attentions in inflated SD's transformer blocks includes Spatial-Temporal Attention, Cross-Frame Gated Attention, and Modulated Cross Attention, whereas Cross-Frame Gated Attention layers are not appended in inflated ControlNet. If a binary mask, the intersection of common outer spaces of target bounding boxes, exists, it is utilized for inpainting before each denoising step. This process helps preserve regions that are not the target of editing (see Sec. 4.4). For brevity, we omitted timestep $t$ in all variables.

## 3.2 INFLATED STABLE DIFFUSION BACKBONE

**Attention Inflation with Spatial-Temporal Self-Attention.** To exploit pretrained SD which is trained without temporal considerations, recent video editing methods (Wu et al., 2022; Qi et al., 2023; Chen et al., 2023) commonly inflate Spatial Self-Attention along the temporal frame axis. In a similar vein, Ground-A-Video enables inter-frame interaction by refactoring the Spatial Self-Attention into **Spatial-Temporal Self-Attention** while retaining the pretrained weights. In specific, for the latent representation $z_t^i$ of source frame $f^i$, query features are derived from spatial features of $z_t^i$, while key and value features are computed from spatial features of concatenated latents $[z_t^1, \ldots, z_t^N]$. Specifically, this process can be written by the following mathematical representation:

$$Q = W^Q \cdot z_t^i, \quad K = W^K \cdot [z_t^1, \ldots, z_t^N], \quad V = W^V \cdot [z_t^1, \ldots, z_t^N],$$

where $W^Q, W^K, W^V$ are projection matrices.

**Per-frame Inversion with Modulated Cross-Attention.** Although model inflation, such as the Spatial-Temporal Attention discussed above or the Sparse-Causal Attention in (Wu et al., 2022), can contribute to preserving global semantic consistency across frames, prior research on the model inflation (Liu et al., 2023) has shown that it adversely degrades the generation quality of the original SD model, leading to imprecise reconstruction. This occurs because Spatial Self-Attention parameters are employed to compute frame correlations, which have not been taught during the pre-training.

Hence, the inflated SD model falls short for the approximate inversion. To mitigate this issue, they fine-tune the attention projection matrices on the input video.

In contrast, our pipeline adopts a per-frame approach to video inversion using the original, non-inflated SD model without any training. In this process, for each of the $N$ frames within the video $f^{1:N}$, we carry out DDIM inversion and null-text inversion (Sec. 2) individually to obtain inverted latents $z_T^{1:N}$ and optimized null-embedding trajectories $\{\varnothing^{1:N}\}_{t=1}^{T}$. The latents $z_T^{1:N}$ are then subjected to Optical flow guided latents smoothing (Sec. 3.4), before proceeding to the denoising process. Additionally, to guide the separately optimized embeddings $\{\varnothing^{1:N}\}_{t=1}^{T}$ to restore the temporal correlation, we reengineer the Cross-Attention mechanism within the transformer blocks of SD that calculates correspondence between latent pixels and context vectors. Specifically, when performing cfg (Eq. equation 4), unlike conditional prediction where context vectors $\tau^{1:N}$ are uniform across frames, unconditional prediction employs unique context vectors $\varnothing^{1:N}$. Since even minor variations in the context vectors across frames adversely impact the global frame consistency (see Fig. 5), we propose reprogramming the Cross-Attention mechanism into **Modulated Cross-Attention**, which can be formulated as $\mathsf{Attn}(Q, K, V) = \mathrm{Softmax}(\frac{QK^T}{\sqrt{d}}V)$ , with

$$Q = W^Q \cdot z_t^i, \quad K = \left\{ \begin{array}{ll} W^K c_t^i & \text{if } cond \\ W^K \left[c_t^1, \ldots, c_t^N\right] & \text{if } uncond \end{array} \right. , V = \left\{ \begin{array}{ll} W^V c_t^i & \text{if } cond \\ W^V \left[c_t^1, \ldots, c_t^N\right] & \text{if } uncond \end{array} \right. . \tag{5}$$

Here, $z_t^i$ denotes a spatial latent feature of frame $f^i$ at timestep $t$, while '*cond*' and '*uncond*' refer to conditional and unconditional predictions, respectively. In the process of computing the unconditional prediction branch of cfg (Eq. (4)), the proposed modulation produces attention maps that correlate with the similarity between $z_t^i$ and the merged null-embeddings $[c_t^1, \ldots, c_t^N]$, thus opening a path for interaction between variant context vectors.

**Video Groundings with Cross-Frame Gated Attention.** Li et al. (2023b) proposed GLIGEN that achieves open-world grounded text-to-image generation via continual learning. In specific, they extend the SD model by adding a gated self-attention layer and fine-tuning this layer to incorporate layout information, while freezing the original SD weights. Ground-A-Video adopts GLIGEN's gated attention module and refactor the operation to accommodate the video-grounding in a time-consistent fashion, which has not been explored before.

Following the same notations, we denote the semantic information associated with the $i$-th grounding entity as $e_i$ and a set of bounding box coordinates for the $i$-th grounding entity as $\boldsymbol{l}_i$. Next, we define the concept of 'grounding' for a single image, expressed as $\boldsymbol{g} = [(e_1, \boldsymbol{l}_1), ..., (e_M, \boldsymbol{l}_M)]$ with $M$ denoting the number of entities to ground. Expanding from image space to video space, we aggregate groundings across $N$ frames of the input video, yielding $\boldsymbol{g}^{1:N} = [\boldsymbol{g}^1, ..., \boldsymbol{g}^N]$. Subsequently, we disentangle the complex editing objectives and introduce handcraft modifications, transitioning $e_i$ to $e_i'$ for $i = 1, \ldots, M$, resulting in $\boldsymbol{g}'^{1:N}$—the revised grounding input for our model. For example, consider the first frame's grounding, denoted as $\boldsymbol{g}^1$, within the input video in Figure 3, which takes the form of $[('rabbit', (0.0, 0.0, 0.7, 0.9)), \ldots, ('table', (0.0, 0.8, 1.0, 1.0))]$. Then, the editing objectives $\Delta\tau = \{\tau_{rabbit} \rightarrow \tau_{kangaroo}, \ldots, \tau_{table} \rightarrow \tau_{snow}\}$ transforms grounding $\boldsymbol{g}^1$ to grounding input $\boldsymbol{g}'^1$, which is written as $[('kangaroo', (0.0, 0.0, 0.7, 0.9)), \ldots, ('snow', (0.0, 0.8, 1.0, 1.0))]$. The prepared grounding input requires post-processing prior to proceeding to the reverse diffusion. The semantic information $e_i^j$ is directed to the CLIP text encoder, where it is transformed into text tokens. Meanwhile, the layout information $l_i^j$ is encoded using Fourier embedding, following Mildenhall et al. (2021), to produce layout tokens. The text tokens and layout tokens are then projected to the single 'grounding' space using an MLP, resulting in grounding tokens:

$$u_i^j = \mathrm{MLP}([\underbrace{\mathcal{E}_{\mathrm{CLIP}}(e_i^j)}_{\text{text token}}, \underbrace{\mathrm{Fourier}(l_i^j)}_{\text{layout token}}]). \tag{6}$$

Given the grounding tokens for frame $i$ and an intermediate latent representation of frame $i$ at time $t$, denoted as $z_t^i$, our goal is to project the grounding information onto the visual latent representation. However, frame-individual projection leads to temporal incoherence as, for example, the same text conditioning for 'kangaroo' is projected differently onto the $i$-th latent $z_t^i$ and the $j$-th latent $z_t^j$ unless $z_t^i$ is identical to $z_t^j$. Therefore, we propose the **Cross-Frame Gated Attention** which globally

integrates grounding features onto the latent representation via $\text{TS}(\text{Softmax}(\frac{QK^T}{\sqrt{d}}V))$ with:

$$Q = W^Q \cdot [z_t^i, U^i], \quad K = W^K \cdot [z_t^1, U^1, \ldots, z_t^N, U^N], \quad V = W^V \cdot [z_t^1, U^1, \ldots, z_t^N, U^N],$$

where $U^i = [u_1^i, \ldots, u_M^i]$ for the grounding tokens from equation 6, and the operation TS denotes the token slicing, which ensures that the output shape of the proposed attention operation remains consistent with the input shape.

### 3.3 INFLATED CONTROLNET

ControlNet (Zhang & Agrawala, 2023) starts with a trainable copy of SD UNet, purposefully designed to complement the SD. The trainable branch is then fine-tuned to accommodate task-specific visual conditions, extending the input of $\epsilon_\theta (z_t, t, c)$ to $(z_t, t, c, d)$, where $d$ represents the additional conditions such as depth maps or edge maps. This combined locked-copy and trainable-copy framework preserves the original synthesis capabilities of SD while enabling precise control over the structural attributes of the generated images.

To incorporate structural guidance into the video generation process, we employ inflated ControlNet architecture and depth condition. Initially, we estimate depth maps from the source video, converting $d$ to $d^{1:N}$, and apply the Self-Attention inflation and Cross-Attention modulation (Sec. 3.2) on ControlNet while retaining the fine-tuned weights. During the denoising stage, the residual latent features $z_{res}^{1:N} \in \mathbb{R}^{N \times h \times w \times c}$ of ControlNet are first scaled by 'ControlNet Scale' hyperparameter, then transmitted to the inflated SD UNet, as illustrated in Figure 3. The scaling parameter regulates the degree of structural preservation between the input and output (see Fig. 8, 26).

---

**Algorithm 1** Optical Flow guided Inverted Latents Smoothing

---

**Require:** Number of frames $N$, Source video frames $f^{1:N}$, Timesteps $T$, Text-Image Diffusion Model (SD), Optical flow estimator (RAFT), Threshold for magnitude difference $M_{thres}$

$z_T^{1:N} \leftarrow \text{DDIM\_INV}(f^{1:N}, T, \text{SD})$      ▷ run inversion (e.g. DDIM inversion)

**for all** $i = 2, 3, ..., N$ **do**

    $map_{opt}^i \leftarrow \text{RAFT}(f^{i-1}, f^i)$      ▷ obtain optical flow map

    $map_{mag}^i \leftarrow \text{normalize}(\|map_{opt}^i\|)$      ▷ compute magnitude map

    $map_{mask}^i \leftarrow map_{mag}^i < M_{thres}$      ▷ obtain binary mask denoting static region

    $map_{mask}^i = \text{downsample}(map_{mask}^i)$      ▷ downsample to latent-level size

    $z_T^i = z_T^{i-1} * map_{mask}^i + z_T^i * (1 - map_{mask}^i)$      ▷ flow guided smoothing

**end for**

**Return** $z_T^{1:N}$      ▷ temporally smoothed latents

---

### 3.4 OPTICAL FLOW GUIDED INVERTED LATENTS SMOOTHING

As a video consists of a temporal series of images with a considerable overlap of nearly identical pixels, contemporary video compression techniques (Hu et al., 2021; 2022) harness motion information to mitigate temporal redundancy instead of saving individual pixels for every frame of a video. Inspired by this, Chen et al. (2023) introduces pixel-level residuals of the source video into the diffusion process, while Hu & Xu (2023) leverages motion prior to prevent the regeneration of redundant areas for frame consistency.

In our framework, latent codes of the input video are individually computed by performing inversion on each frame through the T2I model. Hence, the approach of aggregating these latent representations and directly feeding them into the denoising process is suboptimal with respect to preserving consistency on static regions. Also inspired by the video codecs, we propose to refine the inverted latents, guided by optical flow information extracted from the input video frames, which accurately captures the motion changes across frames. The pseudo algorithm is shown in Algorithm 1.

Specifically, we first acquire an optical flow map $map_{opt}^i$ between the consecutive frames $(f^{i-1}, f^i)$ using an optical flow estimation network. This flow map is represented as a two-channel image, with each channel capturing vertical and horizontal motion movement. Subsequently, we incorporate the two different motion channels into a one-channel magnitude map $map_{mag}^i$ by calculating the Euclidean distance at each pixel location, followed by channel-wise normalizations. The resulting $map_{mag}^i$ represents the comprehensive motion prior between $f^{i-1}$ and $f^i$. After thresholding

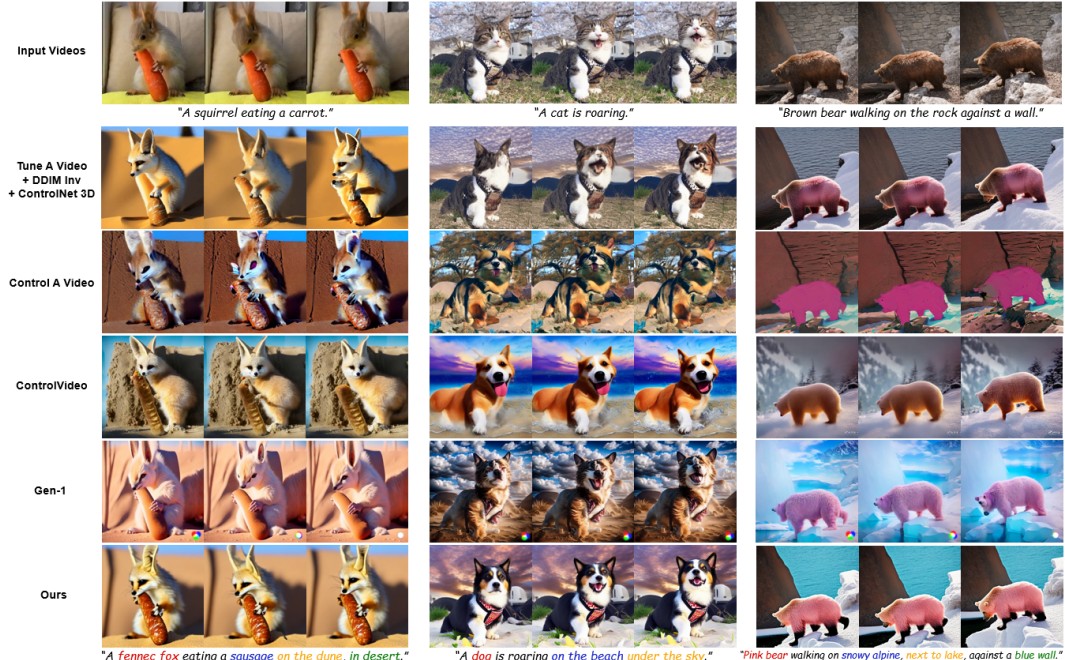

Figure 4: Qualitative comparison with baseline methods: Our results exhibit superior temporal consistency, no mutated body parts, accurate structural preservation, and the highest Edit-Accuracy without omitting or mixing components of edits. Best viewed at high zoom levels.

$map^i_{mag}$ on pre-configured threshold $M_{thres}$, which generates a binary mask ($map^i_{mask}$), we perform smoothing on inverted latents ($z^{i-1}_T, z^i_T$) using the obtained mask. This processing guarantees that static regions share the same pixel values between frames.

## 4 EXPERIMENTS

### 4.1 IMPLEMENTATION DETAILS

We leverage pretrained weights of Stable Diffusion v1.4 (Rombach et al., 2022) and ControlNet-Depth (Zhang & Agrawala, 2023) in addition to self gated attention weights from GLIGEN (Li et al., 2023b). We use a subset of 20 videos from DAVIS dataset (Pont-Tuset et al., 2017). Generated videos are configured to consist of 8 frames, unless explicitly specified, with a uniform resolution of 512x512. We benefit from BLIP-2 (Li et al., 2023a) for the automated generation of video captionings. We then feed-forward video frames and captionings to GLIP (Li et al., 2022) model to obtain bounding boxes of the target objects. For the applications of video style transfer, bounding box coordinates are uniformly set as $[0.0, 0.0, 1.0, 1.0]$ which covers the whole frame. RAFT-Large network (Teed & Deng, 2020) and ZoeDepth (Bhat et al., 2023) are employed for estimating optical flow maps and depth maps, respectively. In the Appendix (Sec. D), we provide a detailed configuration of hyperparameters related to the forward and reverse diffusion processes.

### 4.2 BASELINE COMPARISONS

**Qualitative Evaluation.** We offer a visual comparison against various state-of-the-art video editing approaches in Fig. 4. ControlVideo (CV) (Zhang et al., 2023) stands out as the most relevant work to ours, as it introduces a training-free video editing model that is also conditioned on ControlNet. Control-A-Video (CAV) (Chen et al., 2023) translates a video with ControlNet guidance as well, with a first-frame conditioning strategy. Tune-A-Video (TAV) (Wu et al., 2022) efficiently fine-tunes their inflated SD model on the input video. To ensure a fair evaluation, as TAV is not provided structural guidance, we apply their inflation logic to ControlNet and fine-tune the inflated-SD-ControlNet on the input video. Gen-1 (Esser et al., 2023) presents a video diffusion architecture with additional structure and content guidance specifically designed for video editing. It's worth noting that CAV and Gen-1 train their models with video datasets and that the methods with ControlNet uniformly employed depth guidance for a fair comparison.

**Quantitative Evaluation.** We evaluate the proposed method against aforementioned baselines using both automatic metrics and a user study. The results are outlined in Tab 1.

*(a) Automatic metrics.* We use CLIP (Radford et al., 2021) for automatic metrics. For textual alignment (Hessel et al., 2021), we calculate average cosine similarity between the target prompt and the edited frames. For frame consistency, we compute CLIP image features for all frames in the output video then calculate the average cosine similarity between all pairs of video frames.

| Method | Text-Align | Frame-Con | Edit-Acc | Preserve-Acc | Frame-Con |
|--------|-----------|-----------|----------|--------------|-----------|
| TAV | 0.810 | 0.959 | 2.99 | 3.13 | 3.05 |
| CAV | 0.801 | 0.955 | 2.25 | 2.50 | 2.88 |
| CV | 0.822 | 0.963 | 2.36 | 2.02 | 2.08 |
| Gen-1 | 0.833 | 0.939 | 2.41 | 2.51 | 2.56 |
| Ours | **0.837** | **0.970** | **4.13** | **4.24** | **4.01** |

Table 1: Summary of quantitative evaluations using CLIP (*Left*) and user study (*Right*).

As shown in Tab. 1, our method outperforms baselines in both textual alignment and temporal consistency. *(b) User study.* We surveyed 28 participants to evaluate accuracy of editing and consistency of frames in the edited videos, utilizing a rating scale ranging from 1 to 5. Specifically, to measure editing accuracy, we divided the evaluation into two questions: i."*Were all the elements in the input video that needed to be edited accurately edited?*" ii."*Were the elements in the output video that needed to be preserved accurately preserved?*" Tab. 1 shows that our method surpasses the baselines in all three aspects, particularly with a significant lead in Edit-Accuracy and Preserve-Accuracy.

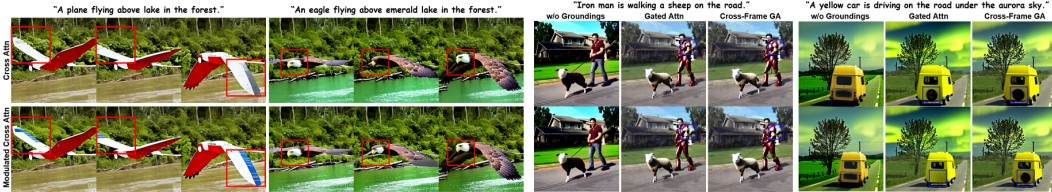

Figure 5: *Left*: Comparison of Modulated Cross-Attention with the original Cross-Attention, in the context of individually optimized null-embeddings. *Right*: Comparison of Cross-Frame Gated Attention against original frame-independent gated mechanism and the absence of a gated attention layer. The rightmost case shows editing with frame-independent groundings can yield subpar results compared to editing without any groundings.

## 4.3 ABLATION STUDIES

**Attentions.** We compare the use of the original Cross-Attention with the Modulated Cross-Attention in Fig. 5-*Left*. The results reveal variations in unconditional context vectors lead to distinct appearances of the subject within a video and the Modulated mechanism promotes the coherency of the subject's appearance. In Fig. 5-*Right*, we compare our proposed Cross-Frame Gated Attention to scenarios with no gated attention (no grounding conditions) and with the direct application

|  | Text-Align | Frame-Con |
|--|-----------|-----------|
| w/o Modulated CA | 0.835 | 0.967 |
| w/o Groundings | 0.802 | 0.960 |
| w/o Cross-Frame GA | 0.829 | 0.956 |
| w/o ControlNet | 0.823 | 0.948 |
| Full components | **0.837** | **0.970** |

Table 2: Quantitative assessments on pipeline components using CLIP.

of GLIGEN's Gated Attention. The example of 'Iron man' illustrates the semantic edit of 'man → Iron man' is neglected in the absence of groundings guidance and that frame-independent Gated Attention causes discrepancy in the appearance of the shoulder area of 'Iron man' The 'yellow car' example reveals that grounding guidance can result in inferior results compared to the generation with no grounding conditions at all, if not applied in a cross-frame manner. These results underscore the pivotal role of Cross-Frame Gated Attention in both edit-accuracy and time-consistency. Moreover, we provide a quantitative analysis detailing the impact of each module in Tab.2.

**ControlNet.** We ablate the utilization of ControlNet-Depth. Fig. 2- and Fig. 6-*Right* depict instances of inaccurate reconstruction of the input video. ControlNet's structure guidance is necessitated to draw accurate structure of 'rabbit' and 'penguin' inside the bounding boxes, respectively. To further validate the usage of ControlNet, we conducted a quantitative analysis to assess the impact of the proposed inflated ControlNet, as presented in Table 2.

**Optical Flow Smoothing.** To assess the impact of optical flow-guided inverted latents smoothing, we ablate the smoothing using three threshold values: 0 (no smoothing applied), 0.2 and 0.6. As revealed in Fig. 6, our flow-guided smoothing effectively eliminates artifacts within static regions and enhance consistencies. To find optimal threshold value, we computed Frame Consistencies using thresholds of 0.2, 0.3, and 0.4. Notably, the 0.2 threshold resulted in superior frame consistency (0.970 for threshold 0.2 > 0.968 for threshold 0.3, 0.964 for threshold 0.4).

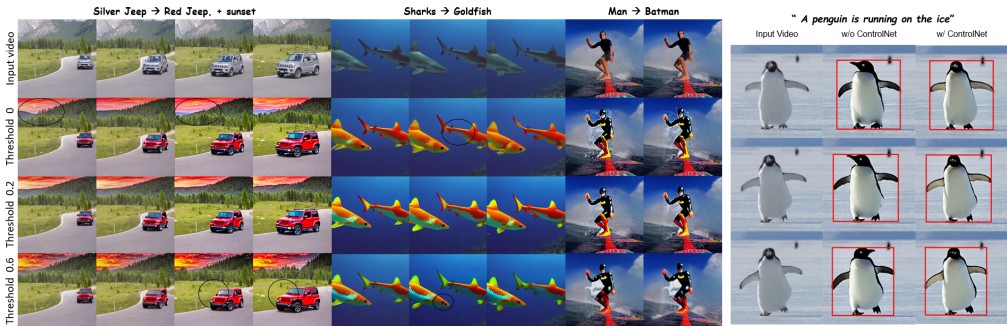

Figure 6: *Left*: Editing outcomes using different thresholds for Optical Flow Smoothing. *Right*: Input video reconstruction results with and without ControlNet guidance.

## 4.4 APPLICATIONS OF GROUND-A-VIDEO

**Groundings-guided Editing with Inpainting.** Employing a grounding condition offers a significant advantage, as it facilitates the creation of an inpainting mask. This mask is readily obtained by intersecting the shared outer areas of bounding boxes. By utilizing this acquired mask during editing, which identifies regions in the source video that should remain unaltered, Preserve-Accuracy is further enhanced. For instance, in Fig. 7-*Middle*, the common outer spaces of red bounding boxes and blue bounding boxes form a binary mask which is used for inpainting. Yet, in Fig. 7-*Bottom*, there is no intersection of common outer spaces, and thus, inpainting is not applied in this scenario.

**Video Style Transfer & Text-to-video Generation with Pose Control.** In the video style transfer task of 7-*Middle*, target style texts are injected to UNet backbone in both Cross-Frame Gated Attention and Modulated Cross Attention layers. The second row showcases the application of style transfer, while the third row shows style transfer combined with attributes editing. Our method adeptly translates input video into the desired style, all while incorporating semantic edits when necessary. Fig. 7-*Right* illustrates the use of Ground-A-Video for zero-shot text-to-video generation with pose map guidance. The pose map images are sourced from Ma et al. (2023). These spatially-continuous pose map conditions are integrated into the diffusion reverse process via inflated ControlNet.

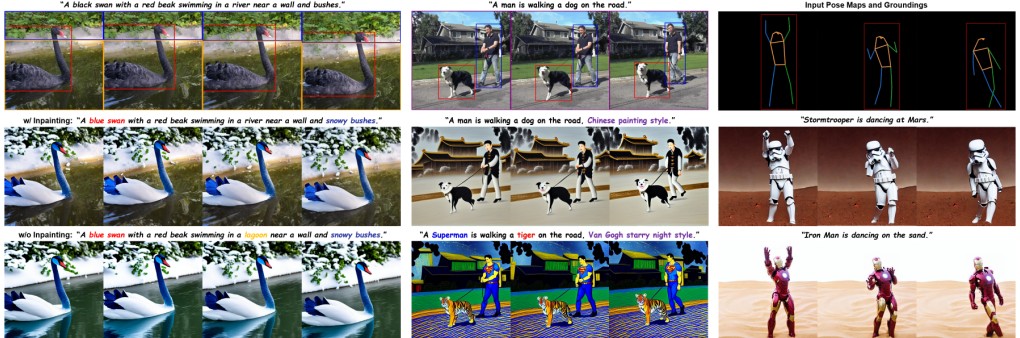

Figure 7: *Left*: Video editing with and without inpainting. *Middle*: Video style transfer with attribute change. *Right*: Text-to-video generation with pose maps. All three results are generated with zero-shot inference.

## 5 CONCLUSION

In our work, we addressed the problem of complex multi-attribute editing of a video and proposed an answer of utilizing both spatially-continuous and discrete conditions. Ground-A-Video offers precise video editing capabilities without the need for fine-tuning off-the-shelf diffusion models on any video data. We demonstrated the power of our method on

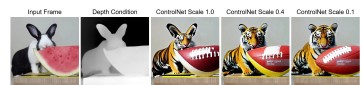

Figure 8: Video editing results with different ControlNet Scales.

various input videos and applications. We provided detailed comparisons to existing baselines along with an extensive user study, demonstrating its superiority in terms of consistency and accuracy.

**Limitations.** Since video groundings play a crucial role in our pipeline, misleading groundings (e.g., incorrect bounding box coordinates) may result in inaccurate editing outcomes. Although the use of ControlNet inherently brings the issue of structural flexibility between the input and output, this can be effectively controlled through the 'ControlNet Scale' hyperparameter (see Fig. 8, 26).

**Ethics & Reproducibility.** The use of T2I foundation models brings forth several ethical considerations. These models possess the potential for malicious applications, such as the creation of misleading or counterfeit content, which could yield adverse societal consequences. Our work heavily relies on one such model, making it susceptible to these concerns. Furthermore, the models were trained on a dataset of internet-sourced images (Schuhmann et al., 2022), which may encompass inappropriate content and inherent biases. Consequently, these models might perpetuate such biases (Mishkin et al., 2022) and generate inappropriate imagery. To address these potential concerns and foster reproducibility, we will release our source code, model, and data under a license that encourages ethical and legal usage. Additional information regarding experiments, implementations and the code base can be found in the Appendix (see Sec. D).

**Acknowledgements.** This research was supported by the National Research Foundation of Korea (NRF) (RS-202300262527), Field-oriented Technology Development Project for Customs Administration funded by the Korean government (the Ministry of Science & ICT and the Korea Customs Service) through the National Research Foundation (NRF) of Korea under Grant NRF2021M3I1A1097910 & NRF2021M3I1A1097938, Korea Medical Device Development Fund grant funded by the Korea government (the Ministry of Science and ICT, the Ministry of Trade, Industry, and Energy, the Ministry of Health & Welfare, the Ministry of Food and Drug Safety) (Project Number: 1711137899, KMDF_PR_20200901_0015), and Culture, Sports, and Tourism R&D Program through the Korea Creative Content Agency grant funded by the Ministry of Culture, Sports and Tourism in 2023.

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

## A    COMPARISON TO PER-FRAME T2I EDITING METHODS

To further underscore the effectiveness and indispensability of Ground-A-Video's pipeline design, we offer comparisons with two groundbreaking T2I editing methods applied in a per-frame manner, i.e., per-frame GLIGEN (Li et al., 2023b) editing and per-frame ControlNet (Zhang & Agrawala, 2023) editing. Note that random seeds were consistently fixed when sampling each frame and DDIM inversion was utilized to give additional compositional guidance for both GLIGEN editing and ControlNet editing. As demonstrated in Fig. 9, a straightforward extension of the T2I methods in a frame-by-frame fashion leads to significant appearance inconsistencies in the video translation task.

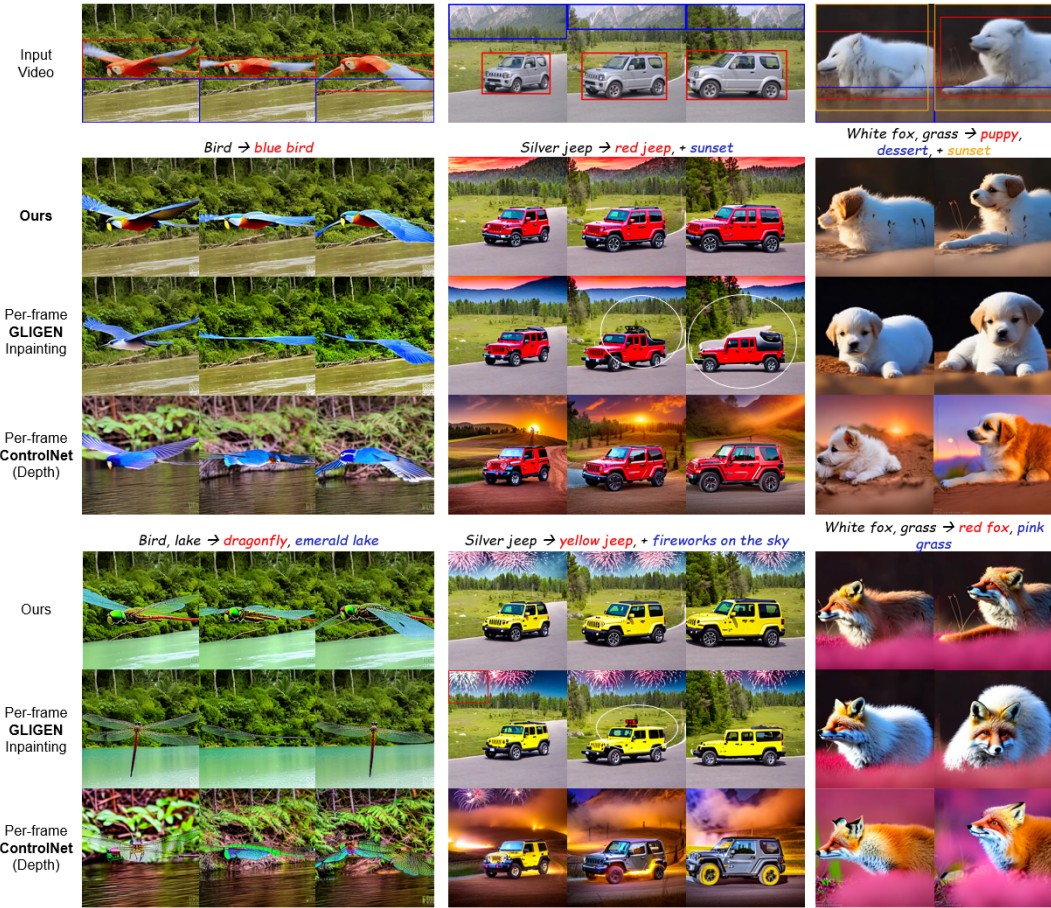

Figure 9: Comparison with GLIGEN inpainting and ControlNet (depth) img2img on a frame-by-frame basis: The direct application of state-of-the-art image editing methods to the video editing task results in a substantial lack of appearance consistency among frames.

## B    UTILIZING OPTICAL FLOW-GUIDED INVERTED LATENTS SMOOTHING IN TUNE-A-VIDEO

To further validate robustness of the proposed optical flow-guided inverted latents smoothing, we demonstrate its application within the Tune-A-Video (TAV) (Wu et al., 2022) framework. Similar to our approach, TAV commences its denoising process from DDIM-inverted latents. Fig. 10 illustrates a comparison between the application of flow smoothing and its absence within the TAV editing framework. Our smoothing technique effectively ensures consistency within static regions (e.g., the avocado on the left and the corn on the right) and eliminates artifacts across frames. It is noteworthy that our proposed optical flow smoothing is applicable to any video editing framework utilizing DDIM inversion.

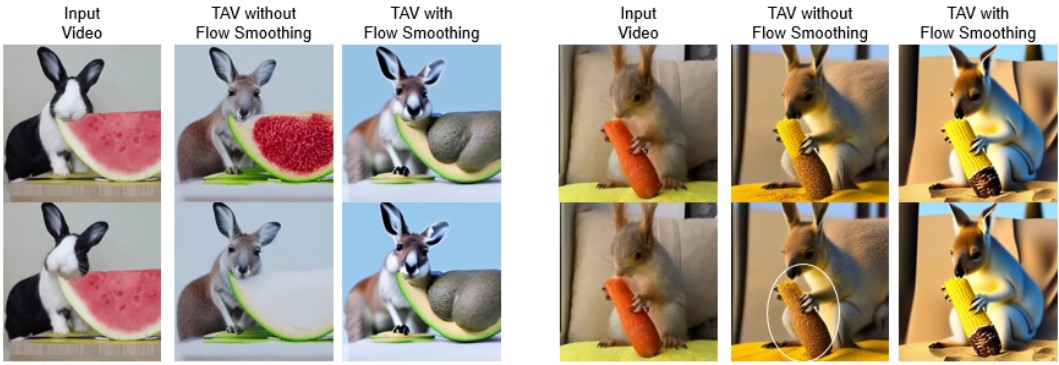

Figure 10: Application of the flow-guided inverted latents smoothing in the Tune-A-Video framework.

## C RELATED WORK

### C.1 CONTROL OVER TEXT-TO-IMAGE GENERATION

Internet-scale datasets of image-text pairs (Schuhmann et al., 2022) have driven remarkable advancements in diffusion models within the realm of text-image generation (Rombach et al., 2022; Ramesh et al., 2022; Saharia et al., 2022). Consequently, the focus of text-image models naturally shifted towards the challenge of controllable generation. Controlled image generation can be achieved through two distinct modalities: spatially-continuous conditions and discrete conditions. Notable contributions in the former include T2I-Adapter (Mou et al., 2023) and ControlNet (Zhang & Agrawala, 2023), which augment pretrained T2I models with auxiliary networks. These networks are specifically trained to produce images conditioned on spatially-continuous visual cues such as depth maps and edge maps. Conversely, GLIGEN (Li et al., 2023b) and Reco (Yang et al., 2023) fine-tune the T2I model to accommodate discrete layout information, comprising bounding box coordinates and textual captions.

Another approach to exert control over the generation process is to heavily reference the input image, with the goal of preserving structural and background attributes. Prompt-to-prompt (Hertz et al., 2022) and Pix2pix-zero (Parmar et al., 2023) achieve structure-preserving image generation by using cross-attention maps of the input image, while Plug-and-play (Tumanyan et al., 2023) and MasaCtrl (Cao et al., 2023) manipulates spatial features of the self-attention mechanism, which are extracted from the input image diffusion trajectory. Alternate strategies (Mokady et al., 2023; Kawar et al., 2023) attain enhanced editing capabilities through optimization techniques. Yet, straightforward application of previously mentioned methods in a frame-by-frame manner for video editing outputs pronounced flickering and temporal inconsistencies.

### C.2 DIFFUSION MODELS FOR VIDEO

When juxtaposed with text-image generation, generating videos in a text-only condition poses a significantly elevated challenge due to the complexity of constraining temporal consistency along with the scarcity of extensive text-video datasets, which are both resource-unfriendly. Video Diffusion Models (VDM) (Ho et al., 2022b) designs a space-time factorized 3D UNet architecture trained on both image and video modalities. ImagenVideo (Ho et al., 2022a) trains cascaded VDMs with v-prediction parameterization. Make-A-Video (Singer et al., 2022) and MagicVideo (Zhou et al., 2022) adopt a similar approach in that they train T2V models transferring from pretrained T2I models. More recently, Text2Video-Zero (Khachatryan et al., 2023b) achieves zero-shot text-to-video generation by enhancing the latent features of the generated frames with motion dynamics and restructuring the Spatial Self-Attention along the frame-axis.

Simultaneously, significant research efforts have been dedicated to video editing tasks (Xing et al., 2023). Pioneering work in this field, exemplified by Tune-A-Video (Wu et al., 2022), has employed the approach of fine-tuning query projection matrices in attention layers to effectively retain information from the source video. Video-P2P (Liu et al., 2023) adopts an approach of fine-tuning a Text-to-Set model on the source video as well, additionally introducing a decoupled-guidance atten-

tion control for the inference stage. More recent one-shot video editing frameworks include Motion Director (Zhao et al., 2023) and VMC (Jeong et al., 2023), where they both aim to customize motion patterns presented in the source video. To eliminate the necessity of optimizing model weights entirely, various zero-shot editing techniques have been introduced. Vid2vid-zero (Wang et al., 2023) focus on a cross-attention maps guidance while Pix2Video (Ceylan et al., 2023) employs a self-attention maps injection mechanism, where the attention maps are obtained in the input video inversion stage in both methods. It's worth noting these two distinct attention control methods are orthogonal and thus can be utilized at the same time. Furthermore, FateZero (Qi et al., 2023) introduces a training-free video editing pipeline that utilizes cross-attention maps from the input video inversion trajectory. These maps are used to calculate blending masks, aiming to enhance consistency in the background regions of the generated videos. In contrast to approaches that involve no training on video or training on a single video, Gen-1 (Esser et al., 2023) and Control-A-Video (Chen et al., 2023) adopt a training-based approach for video translation, incorporating additional structural guidance. Notably, Control-A-Video achieves efficient convergence through their innovative first-frame conditioning method, optimizing resource utilization. Recent methodologies (Chu et al., 2023; Hu & Xu, 2023) have made significant advancements in frame consistency by integrating an optical flow warping mechanism into their generation process. A promising concurrent work, TokenFlow (Geyer et al., 2023), accomplishes time-consistent video editing that strongly preserves spatial layout by enforcing consistency on the internal diffusion features across frames during the denoising process.

## D  EXPERIMENTAL DETAILS AND IMPLEMENTATIONS

In this section, we provide experimental details of our method and the compared baselines. **Ground-A-Video**(Depth) utilizes DDIM inversion then performs null-text optimization with the default hyperparameters in Mokady et al. (2023). In the flow-driven inverted latents smoothing stage, the magnitude threshold $M_{thres}$ is set to 0.2. At inference, DDIM scheduler (Song et al., 2020a) with 50 steps and classifier-free guidance (Ho & Salimans, 2022) of 12.5 scale is used. The executable code base will be available at our project's repository.

For the comparison with **Tune-A-Video-Control**(Depth) (Wu et al., 2022), we utilize the authors' provided implementation[1] and also adjust ControlNet (Zhang & Agrawala, 2023) following their inflation guidelines. For fine-tuning, we train the model on each example video for 500 iterations using their default parameters. During inference, we utilize DDIM inversion with 50 steps and employ diffusion reverse with the DDIM scheduler, also using 50 steps. The model architecture and training script for ControlNet-attached Tune-A-Video will also be made available in our project's code repository. For the comparison with **Control-A-Video**(Depth) (Chen et al., 2023), we utilize their official implementation[2] with the following hyperparameter settings: 50 steps for inference, a video scale of 1.5, and a noise threshold of 0.1. In our comparison with **ControlVideo**(Depth) (Zhang et al., 2023), we leverage their open-source code repository[3] and demo website[4]. We perform inference using 50 steps, and we apply interleaved-frame smoothing with steps set at [19, 20]. Although Gen-2, the successor to **Gen-1** (Esser et al., 2023), has been recently announced, it is important to note that Gen-2 supports sole text-to-video generation functionality, which differs from the focus of our paper, which is centered on comparing video editing performance. Thus, we have utilized Gen-1 for our comparisons and the results from Gen-1 were generated by executing its web-based product.

---

[1] https://github.com/showlab/Tune-A-Video
[2] https://github.com/Weifeng-Chen/control-a-video
[3] https://github.com/YBYBZhang/ControlVideo
[4] https://replicate.com/cjwbw/controlvideo

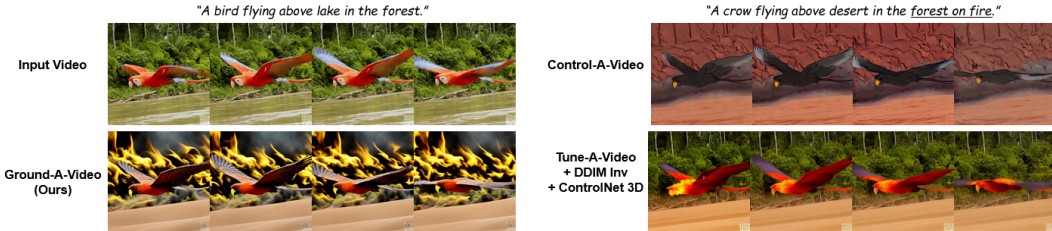

Figure 11: *Left*: Input video and Ground-A-Video's editing result, driven by the target prompt "*A crow flying above desert in the forest on fire.*" *Right*: Cases of **Neglected Edit** and **Edit on Wrong Element**. The editings are driven by the same target prompt, using Control-A-Video and Tune-A-Video-ControlNet, respectively.

## E  SEMANTIC MISALIGNMENT & ADDITIONAL COMPARISONS

In our study, we define the concept of semantic misalignment within the realm of video editing. To facilitate better understanding, we offer illustrative examples of various forms of semantic misalignment through baseline comparisons.

The most common types of semantic misalignment encountered in video editing are '**Neglected Edit**' and '**Edit on Wrong Element**'. Neglected Edit is signified by the omission of one of the intended attribute edits, while Edit on Wrong Element refers to cases where a semantic edit is applied to an element that was not the intended target of the edit. These misalignments are typically observed in scenarios where a wide range of different edits need to be performed. For instance, as shown in Fig. 11, editing scenario can be represented as $\{\tau_{\text{bird}} \rightarrow \tau_{\text{crow}}, \tau_{\text{lake}} \rightarrow \tau_{\text{desert}}, \tau_{\text{forest}} \rightarrow \tau_{\text{on fire}}\}$. In the first row of Fig. 11-*right*, the edit of '$\tau_{\text{forest}} \rightarrow \tau_{\text{on fire}}$' is omitted in the results. The results in the second row provide an example of Edit on Wrong Element, specifically visualizing the occurrence of '$\tau_{\text{bird}} \rightarrow \tau_{\text{on fire}}$. The next type is '**Mixed edit**', which denotes the situations where two or more separate edits become intertwined and mutually affect each other, straying from the original intention. This issue is commonly observed in complex editing scenarios that involve object color changes. For instance, in the second row of Fig. 12-*left*, the edits '$\tau_{\text{car}} \rightarrow \tau_{\text{red}}$' and '$\tau_{\varnothing} \rightarrow \tau_{\text{sunset}}$' become intertwined as '$\tau_{\text{red}} \cdot \tau_{\text{sunset}}$' and are applied globally to the entire image within a video. Finally, we address '**Preservation Failure**', which refers to instances where the generated results fail to preserve the regions that are not the target of editing. This issue is not exclusive to complex editing scenarios. For example, in the second row of Fig. 12-*right*, the edit '$\tau_{\text{penguin}} \rightarrow \tau_{\text{kungfu panda}}$' on the foreground object has been successfully performed, but it has failed to preserve the background of the source video.

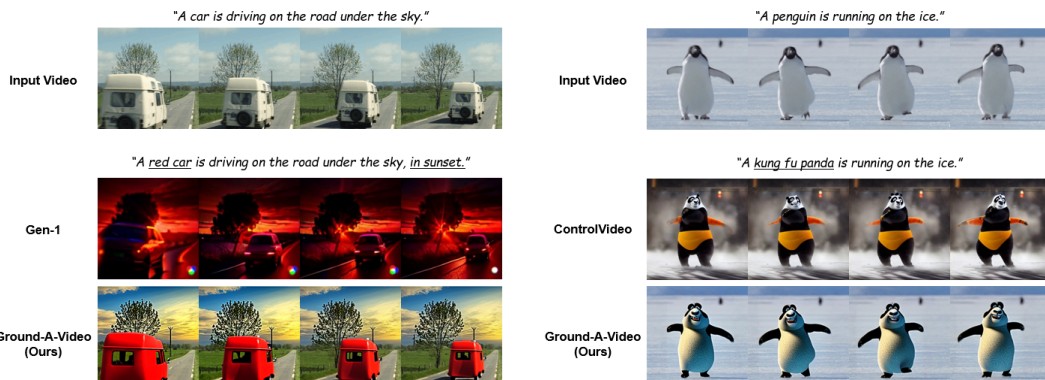

Figure 12: *Left*: An example of **Mixed Edit**.     *Right*: An example of **Preservation Failure**.

# F  SPATIAL CONDITIONS

## F.1  DISCRETE AND CONTINUOUS CONDITIONS

Ground-A-Video framework categorizes spatial conditions into two types: **Spatially-discrete conditions** and **Spatially-continuous conditions**. Spatially-discrete conditions refer to modalities that do not rigidly dictate the precise spatial arrangement of objects or elements within the ultimate output image or video. In our work, we utilize a blend of bounding box coordinates and textual descriptions that pertain to the grounded entity. This spatially-discrete condition doesn't enforce particular layout or structural constraints within the bounding box but rather determines the positions of the entities enclosed by the bounding box within the image frame. In contrast, spatially-continuous conditions provide information that explicitly influences the placement, structure, or spatial distribution of objects throughout every part of the frame, e.g., edge maps and depth maps. To enhance understanding of these terms, we visualize examples of each modalities that are used in our editing pipeline in Fig. 13.

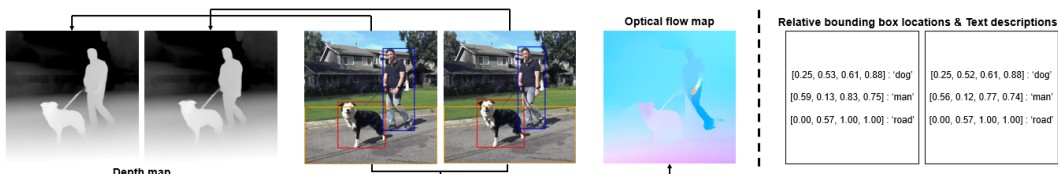

Figure 13: *Left*: Spatially-continuous conditions. *Right*: Spatially-discrete conditions.

## F.2  STATIC AND DYNAMIC GROUNDINGS

Video groundings play a central role in our video editing pipeline, and we would like to classify them into two separate categories based on the video's main subject characteristics: **Static groundings** and **Dynamic groundings**. Static groundings refer to groundings that describe a subject that exhibit minimal movement or remain in a fixed region of the layout. In contrast, dynamic groundings involve groundings describing a subject that transition from one region to another within a video. In Fig. 14, the first row presents an input video featuring a static 'penguin' subject. The subsequent two rows below the first row illustrate our method's successful editing using static groundings associated with the stationary subject. The first row of Fig.15 showcases a video with a dynamic 'car' subject, and the two rows following it demonstrate our method's effectiveness in editing the dynamic subject with the assistance of dynamic groundings.

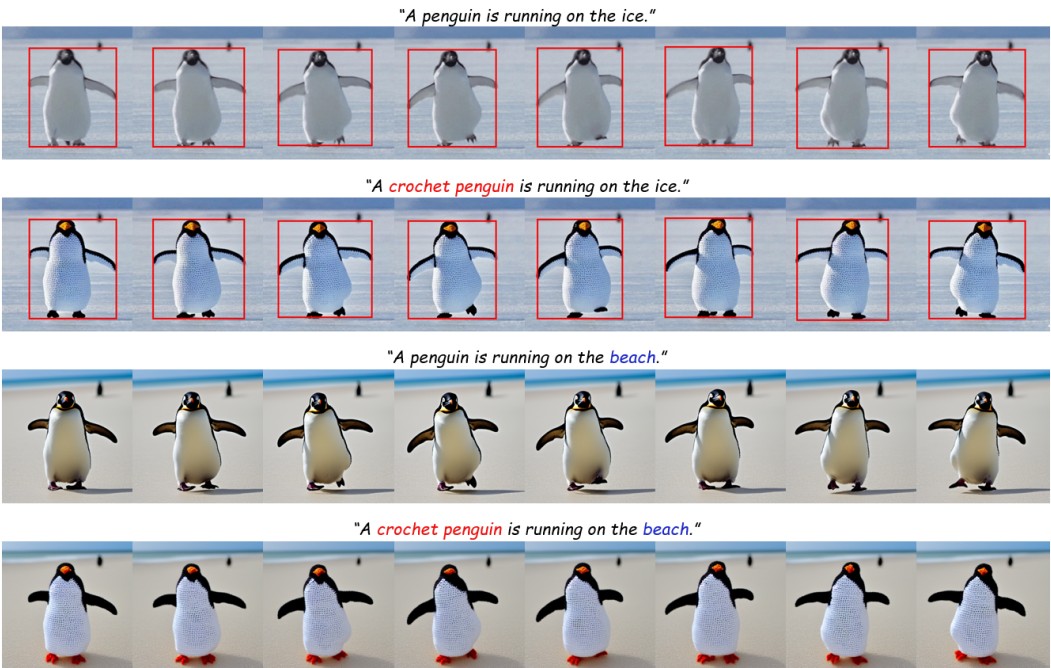

Figure 14: Input video of static subject 'penguin' and static groundings-driven video editing results.

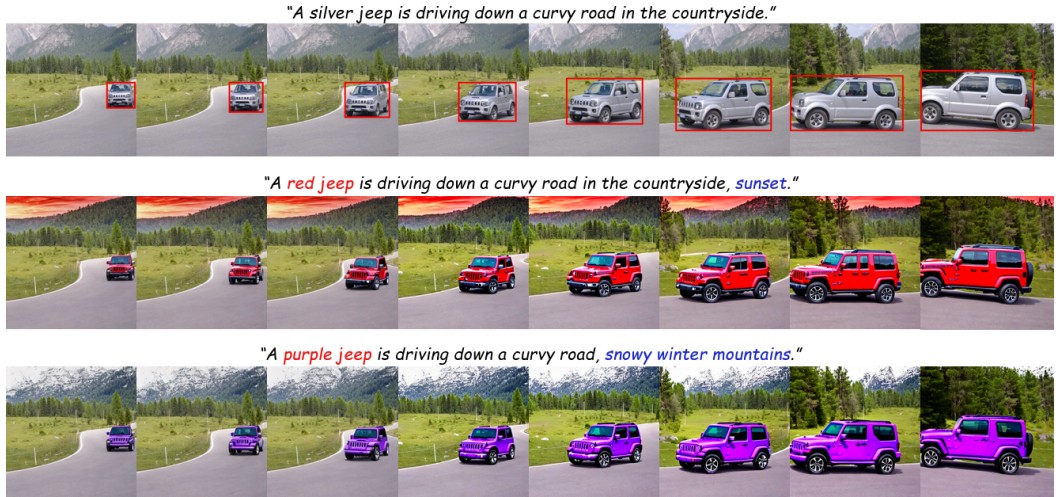

Figure 15: Input video of dynamic subject 'car' and dynamic groundings-driven video editing results.

# G  FULL-LENGTH ADDITIONAL RESULTS

## G.1  VIDEO STYLE TRANSFER

Ground-A-Video is primarily designed for multi-attribute video editing but can also perform video style transfer tasks with temporal consistency. Additional video style transfer applications on various videos are presented in Fig. 16.

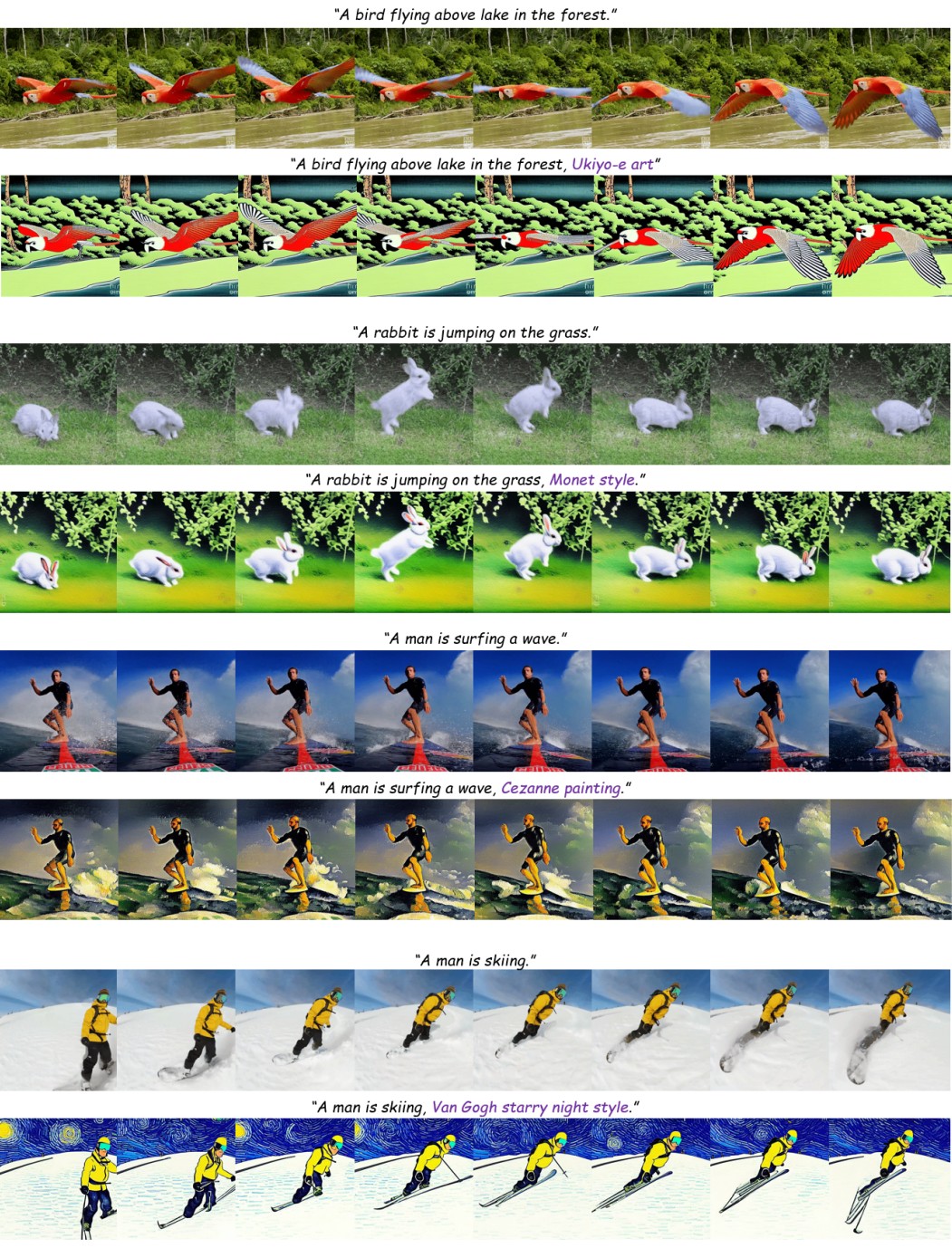

Figure 16: Full-length Video Style Transfer results.

## G.2    MULTI-ATTRIBUTE EDITING

We present additional qualitative results of Ground-A-Video on multi-attribute editing. In each of the Fig. 17, 18, 19, and 20, the top rows visualize[5] the optical flow maps estimated from an input video. The second row shows the input frames, each with annotated bounding boxes around the entities. Our optical flow estimation network, RAFT (Teed & Deng, 2020), requires consecutive images for accurate estimation. Consequently, the first frame, lacking a previous frame for comparison, does not yield any estimation results. For Fig. 21, 22, 23, 24, and 25, the top two rows visualize both the estimated optical flow maps and the corresponding magnitude maps. An optical flow map comprises two-channel images, with each channel representing vertical and horizontal motion residuals. The magnitude map, which combines motion information from both directions into a single-channel image, is computed using $\mathrm{norm}_{\max}(\|map_{opt}\|)$. The normalization is performed along the frame dimension. In the examples, the visualization of magnitude maps demonstrates a successful comprehension of vertical and horizontal movements. On all Fig. 17, 18, 19, 21, 22, 23, 24, 25 under the input video, we present the editing results in ascending order based on the 'number of attributes to be changed' (from small changes to a large number of changes). Furthermore, Fig. 26 presents the editing outcomes using various settings of the ControlNet Scale hyperparameter. It's noteworthy that a reduction in the scaling parameter leads to greater independence in the shapes of the tiger (particularly its ears) and the football from the input video and depth maps. To better assess temporal consistency, full-length results are provided for all figures.

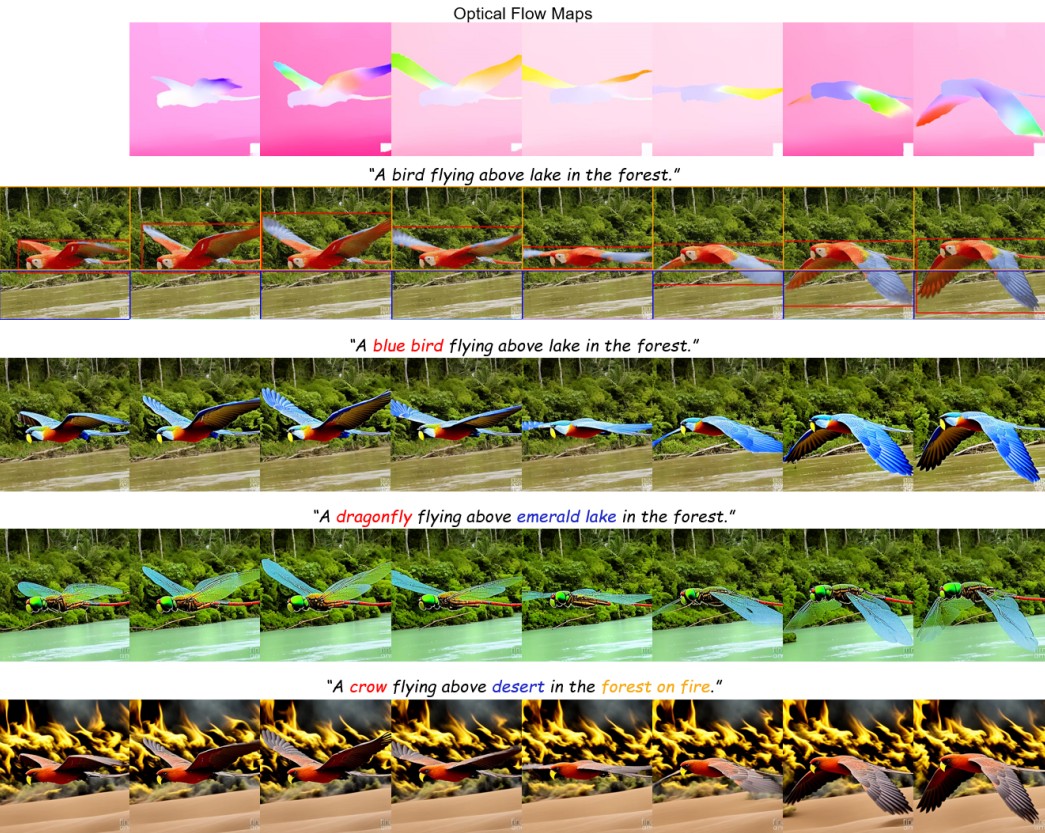

Figure 17: Various editing results on the video of *"A bird flying above lake in the forest."*

---

[5]https://pytorch.org/vision/stable/generated/torchvision.utils.flow_to_image.html

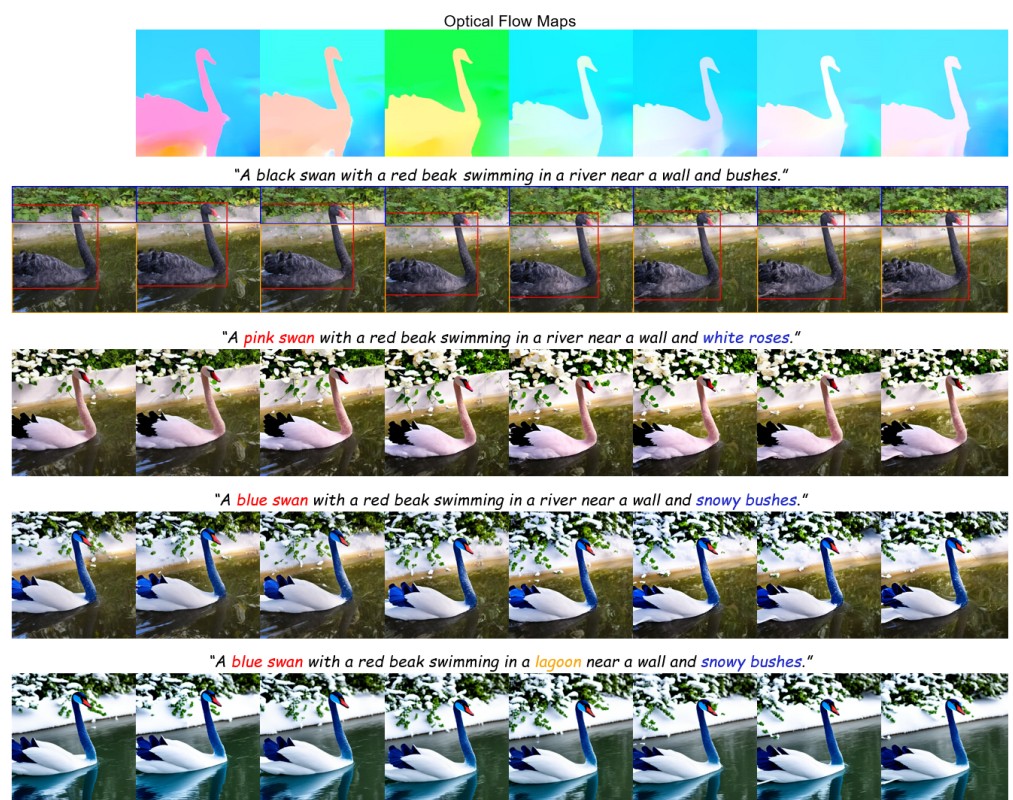

Figure 18: Various editing results on the video of *"A black swan with a red beak swimming in a river near a wall and bushes."*

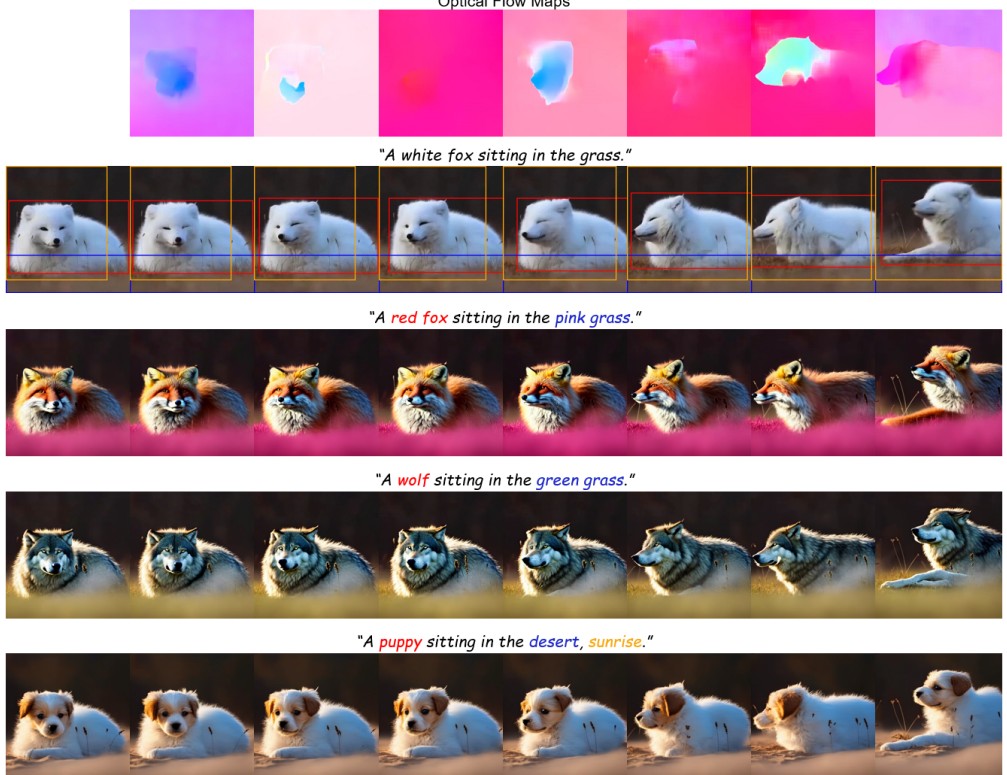

Figure 19: Various editing results on the video of *"A white fox sitting in the grass."*

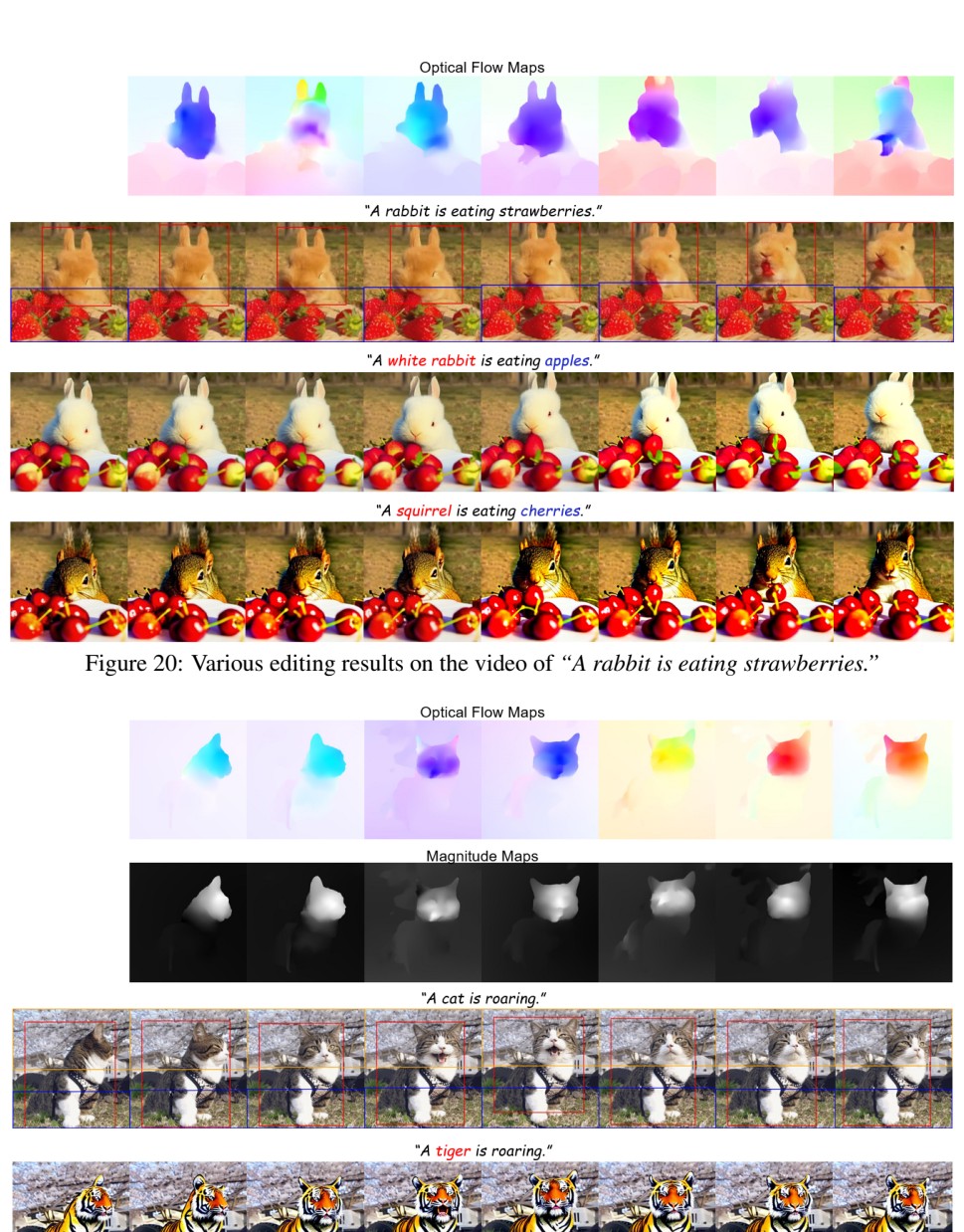

Figure 20: Various editing results on the video of *"A rabbit is eating strawberries."*

Figure 21: Various editing results on the video of *"A cat is roaring."*

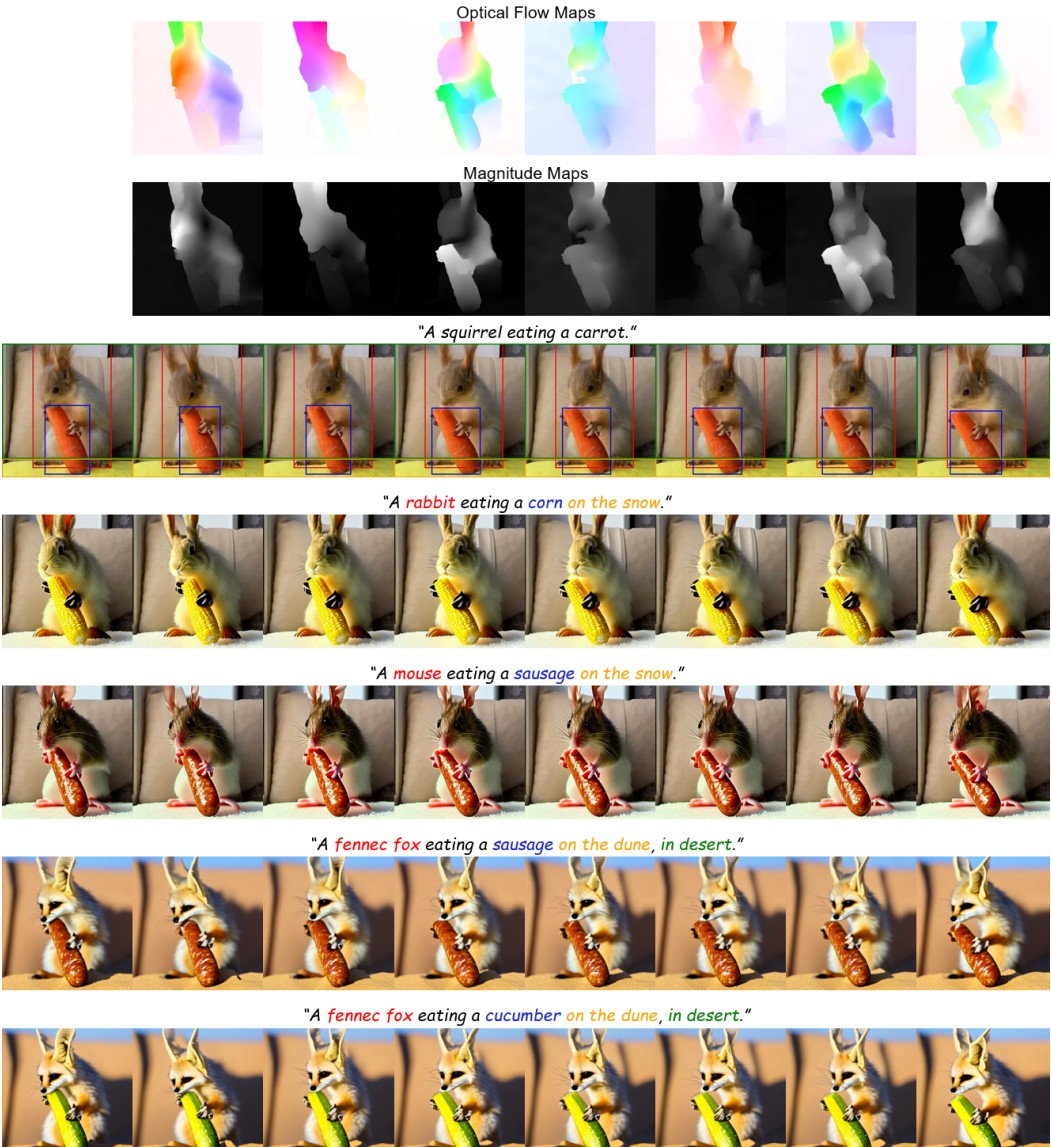

Figure 22: Various editing results on the video of *"A squirrel eating a carrot."*

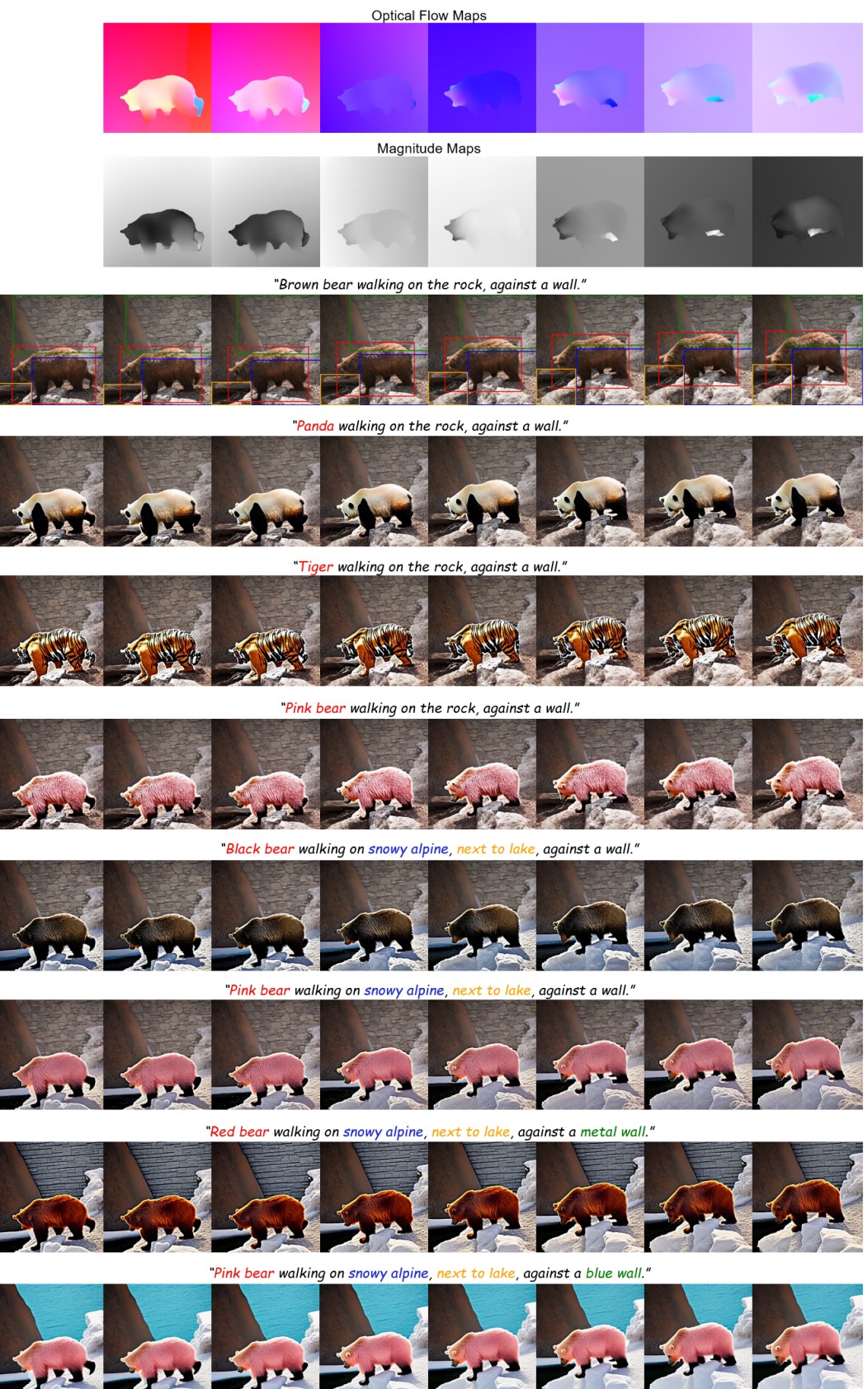

Figure 23: Various editing results on the video of *"Brown bear walking on the rock, against a wall."*

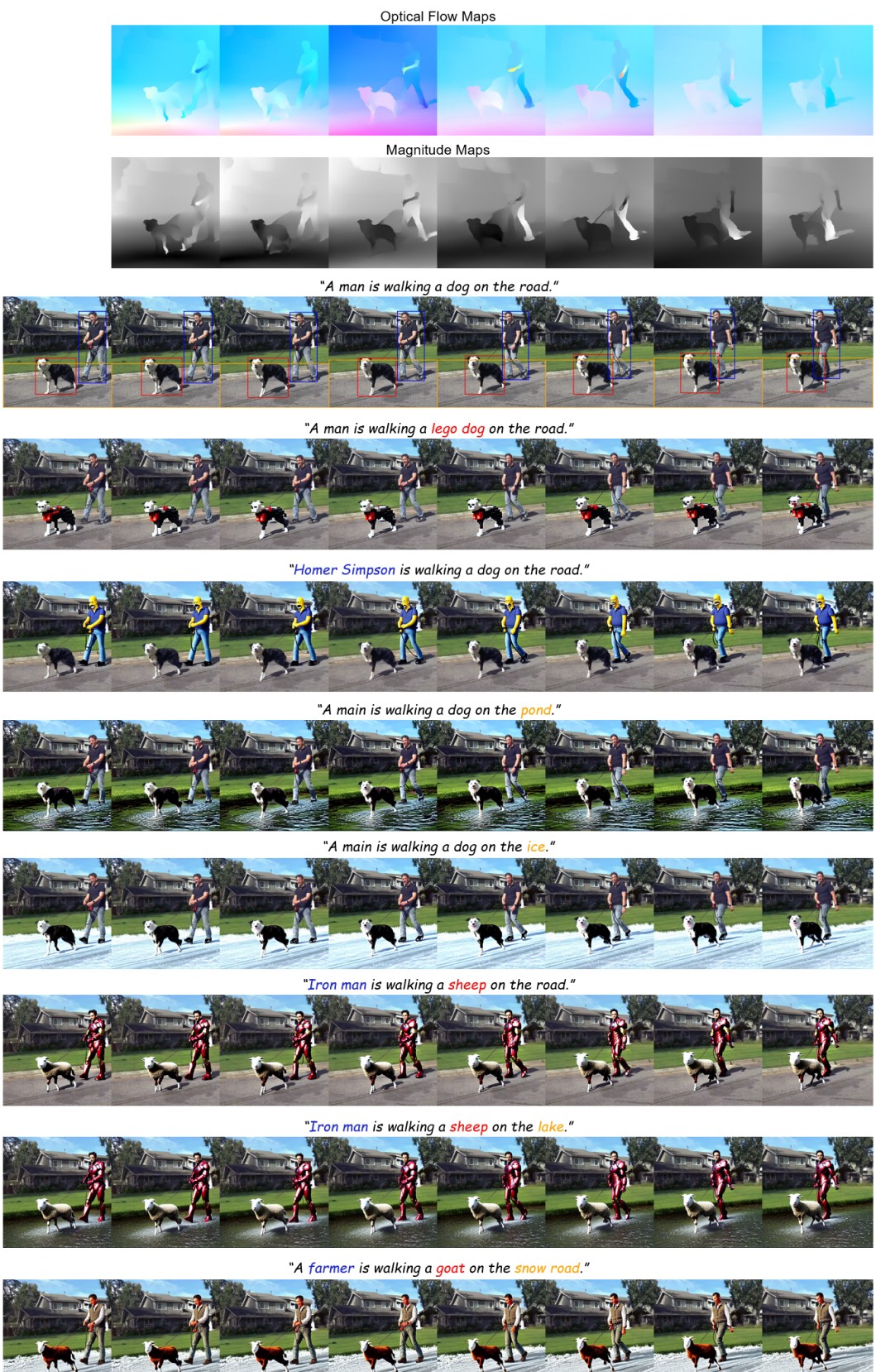

Figure 24: Various editing results on the video of *"A man is walking a dog on the road."*

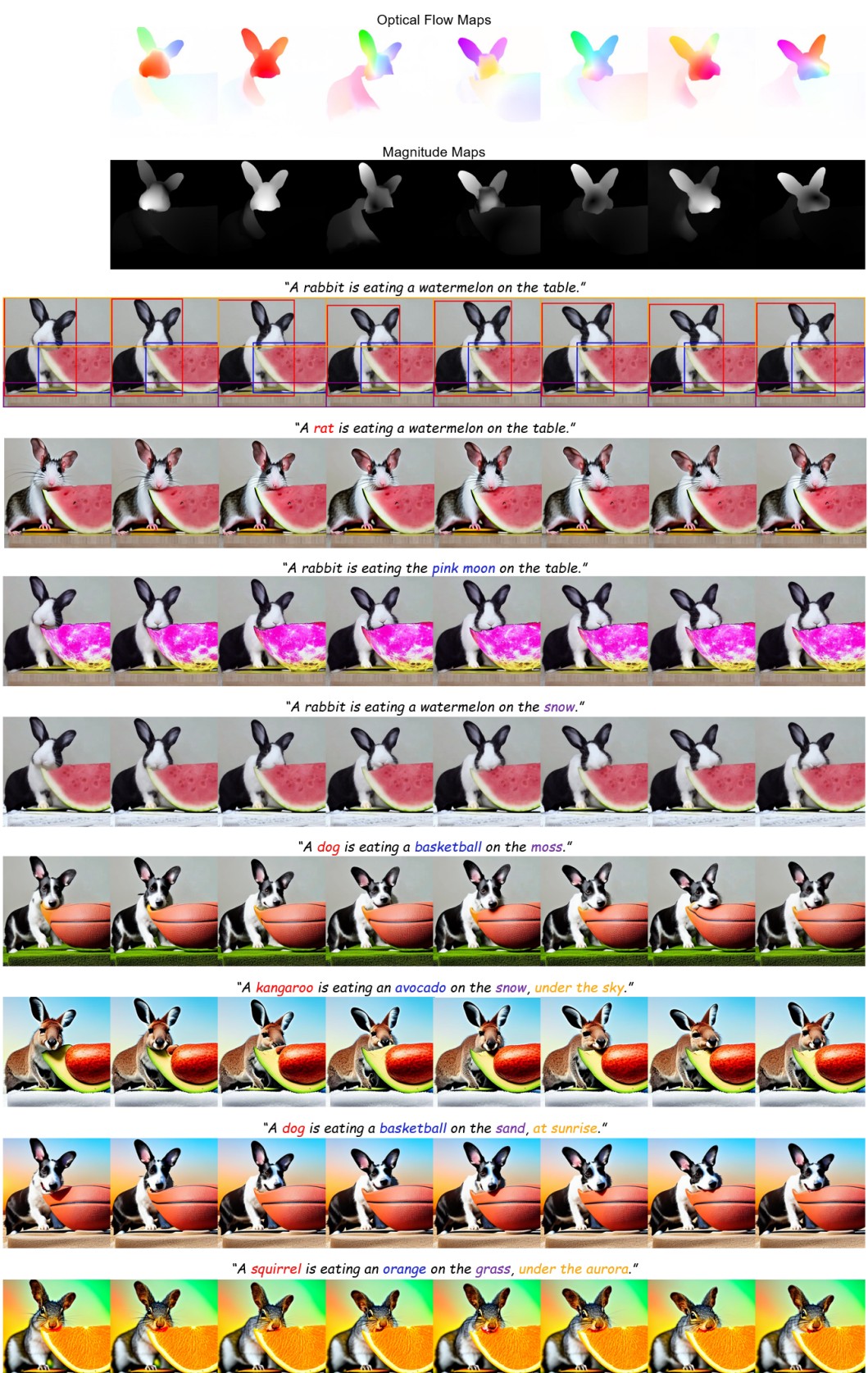

Figure 25: Various editing results on the video of *"A rabbit is eating a watermelon on the table."*

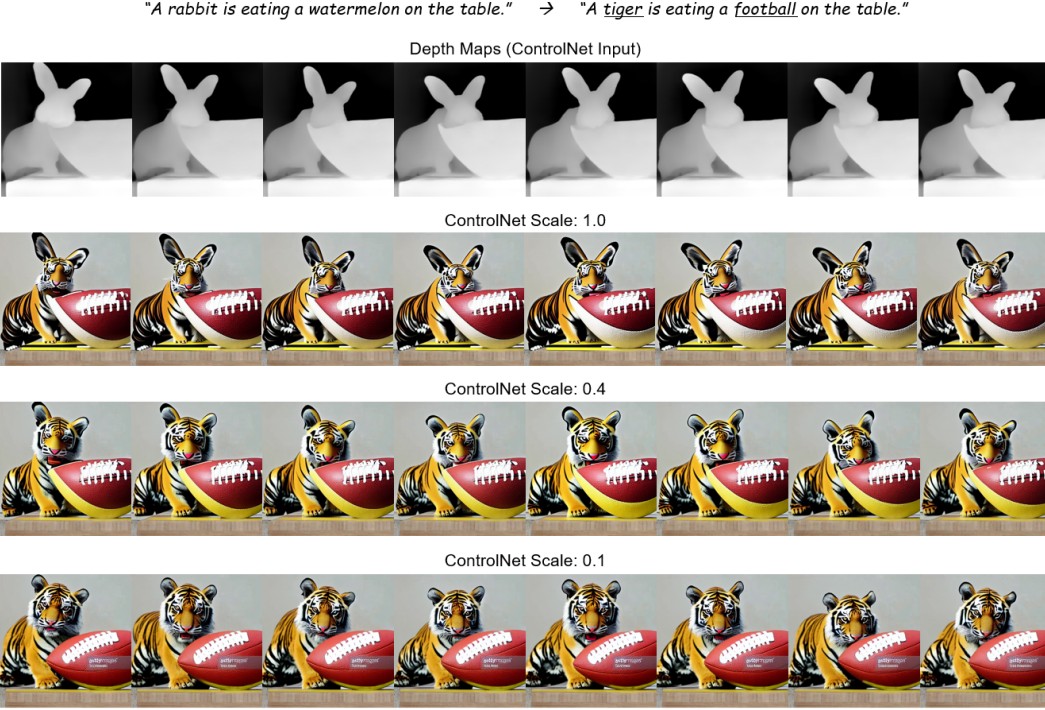

Figure 26: Full frame editing result with varying ControlNet Scales.

