# OpenReview forum: "Ground-A-Video: Zero-shot Grounded Video Editing using Text-to-image Diffusion Models"
_ICLR.cc/2024/Conference — ICLR 2024 poster_

### Official Review · Reviewer_NYGw · 2023-10-28

**Soundness:** 2 fair
**Presentation:** 2 fair
**Contribution:** 2 fair
**Rating:** 6
**Confidence:** 4

**Summary:**

this paper aims to address the challenge of multi-attribute editing in video editing. By incorporating the outputs of grounding models like GLIP and employing a designed attention mechanism, the proposed approach enables precise and temporally consistent video attribute editing. The authors conducted comparative experiments with recent methods, demonstrating superior performance.

**Strengths:**

1. This paper addresses a highly significant problem of achieving consistent fine-grained attribute editing in videos. The introduction of grounding conditions proves to be a direct and effective approach.
2. In addition to the design of grounding conditions, the authors propose mechanisms such as noise smoothing and cross-attention.

**Weaknesses:**

This paper lacks substantial technical innovation as its main contribution lies in expanding the experiments on the input conditions, whether it is the depth or grounding results. The approach used for injecting conditions is based on ControlNet or cross-attention, which is a common practice in stable diffusion and related applications of ControlNet.

**Questions:**

1. The authors provided a visual comparison of the flow smooth effect in Figure 6, but it only includes a single example. Are there additional examples and comparisons with baseline and other methods (excluding flow smooth) demonstrating their effects under the same prompt?
2. during the smoothing process, thresholds are introduced. Are there objective metric comparisons for different thresholds and experiments to evaluate the robustness of the thresholds?

---

> ### Author Response · Authors · 2023-11-19
> **Reply to Reviewer NYGw**
>
> We appreciate the reviewer for the thoughtful comments.
>
> ---
> >**W1.** This paper lacks substantial technical innovation as its main contribution lies in expanding the experiments on the input conditions, whether it is the depth or grounding results. The approach used for injecting conditions is based on ControlNet or cross-attention, which is a common practice in stable diffusion and related applications of ControlNet.
>
> We sincerely thank the reviewer for the insightful review, but gracefully disagree with you in terms of importance of our contribution.
>
> One of the key objectives of our work is to achieve multi-attribute video editing in a '*training-free*' manner. To accomplish this, we leverage the foundational Text-to-Image (T2I) model, Stable Diffusion [1].
> Consequently, we opt for input conditions commonly used in image editing methods, where all input conditions are derived towards the goal of attaining temporal consistency. For example, the groundings are incorporated by our proposed Cross-Frame Gated Attention to assure groundings of each frame contribute to the global appearance consistency across frames within the generated video.
>
> Additionally, in response to concerns that our work may be perceived as an extension of Stable Diffusion or ControlNet-based input conditions, we have added *comparisons with straightforward (per-frame) applications of T2I editing methods* in **Figure 9**.
> As the results demonstrate, injecting input conditions directly leads to significant appearance inconsistencies across generated frames, even when text-image alignment is maintained on individual frames.
> We believe that our methods (Cross-Frame Gated Attention, Inflated ControlNet, Modulated Cross-Attention, Optical Flow guided Latents Smoothing) to solve these inconsistencies are important contribution to the video generative research community, where temporal consistency is the most critical metric in the tasks.
>
> Lastly, we wish to emphasize the novelty of the Modulated Cross-Attention mechanism, which enables interactions between context vectors across frames. This approach has not been explored previously even in a similar context.
>
> ---
> >**W2.** The authors provided a visual comparison of the flow smooth effect in Figure 6, but it only includes a single example. Are there additional examples and comparisons with baseline and other methods (excluding flow smooth) demonstrating their effects under the same prompt?
>
> In response to the reviewer's concern, we have incorporated additional examples showcasing the proposed optical flow smoothing in **Figure 6 (Left)**.
> While we did not identify any prior work specifically addressing the smoothing of inverted latents before the denoising process, we have demonstrated the effectiveness of our flow smoothing by applying it within the Tune-A-Video [3] framework, as depicted in the newly added **Figure 10**.
> In comparison to scenarios where flow smoothing is absent, the application of this technique results in improved consistencies within static regions and the elimination of artifacts.
>
> We also kindly remind the reviewer that our proposed smoothing is applicable to any video translation framework that initiates denoising process from inverted latents.
>
> ---
> >**W3.** during the smoothing process, thresholds are introduced. Are there objective metric comparisons for different thresholds and experiments to evaluate the robustness of the thresholds?
>
> To assess the robustness of the thresholds, we have included a quantitative analysis in the Optical Flow Smoothing Subsection of **Section 4.3**.
> Through empirical experimentation, we determined that the optimal threshold for smoothing falls within the range of 0.1 to 0.5 and the algorithm is quite robust within this ranges. Consequently, in our quest for the ideal value, we computed frame consistencies using thresholds of 0.2, 0.3, and 0.4 with the same video dataset employed in our experiments.
>
> | Threshold | 0.2   | 0.3   | 0.4   |
> |-----------|-------|-------|-------|
> | Frame-Con | 0.970 | 0.968 | 0.964 |
>
> ---
> [1] Rombach, Robin, et al. High-resolution image synthesis with latent diffusion models. In CVPR, 2022.
>
> [2] Zhang, Lvmin, et al. Adding conditional control to text-to-image diffusion models. In ICCV, 2023.
>
> [3] Wu, Jay Zhangjie, et al. Tune-a-video: One-shot tuning of image diffusion models for text-to-video generation. In ICCV, 2023.

---

> ### Author Response · Authors · 2023-11-21
> **Dear Reviewer NYGw**
>
> As the deadline for the Reviewer-Author discussion phase is fast approaching (there is only a day left), we respectfully ask whether we have addressed your questions and concerns adequately.

---

> ### Author Response · Authors · 2023-11-23
> **[Reminder] Summarization of our rebuttal**
>
> Dear reviewer NYGw,
>
> We believe that we have addressed the concerns that you have raised. Specifically,
>
> 1. We clarified our pipeline design and its components, aimed at achieving *training-free* multi-attribute *video* editing.
> To accomplish this goal, we leverage the pretrained Stable Diffusion model [1], employ input conditions used in image editing methods, and repurpose these conditions, all with the important aim of *ensuring temporal consistency in the generated videos*.
>
> 2. To further address your concern that our work may be perceived as an extension of Stable Diffusion or ControlNet [2] -based input conditions, we have added comparisons with straightforward (per-frame) applications of T2I editing methods in **Figure 9**.
> As the results illustrate, direct injection of input conditions leads to significant appearance inconsistencies across generated frames, even when maintaining text-image alignment on individual frames.
> We firmly believe that our proposed methods (Cross-Frame Gated Attention, Inflated ControlNet, Modulated Cross-Attention, Optical Flow guided Latents Smoothing) to solve these inconsistencies are valuable contributions to the field of video generative research, especially in the context where temporal consistency is of paramount importance.
>
>
> 3. In response to the reviewer's concern regarding our proposed Optical Flow Smoothing, we have incorporated additional examples showcasing the proposed algorithm in **Figure 6 (Left)**.
>
>
> 4. To the best of our knowledge, no prior work has attempted any form of smoothing operations on inverted latent representations prior to the denoising process.
> However, in response to the reviewer's request for further validation of our proposed smoothing, we have showcased the effectiveness of our flow smoothing by integrating it into the Tune-A-Video [3] framework, as depicted in the newly introduced **Figure 10**.
> Comparing scenarios with and without flow smoothing, the application of this technique leads to enhanced consistencies within static regions and the removal of artifacts.
>
> 5. To illustrate the robustness of our Optical Flow Smoothing threshold, we have introduced a quantitative analysis in the Optical Flow Smoothing subsection of **Section 4.3**.
> Through empirical experimentation, we have determined that the optimal smoothing threshold lies within the range of 0.1 to 0.5, and the algorithm exhibits robustness within this range. In pursuit of the ideal threshold value, we computed frame consistencies using thresholds of 0.2, 0.3, and 0.4 with the same video dataset used in our experiments.
>
> [1] Rombach, Robin, et al. High-resolution image synthesis with latent diffusion models. In CVPR, 2022.
>
> [2] Zhang, Lvmin, et al. Adding conditional control to text-to-image diffusion models. In ICCV, 2023.
>
> [3] Wu, Jay Zhangjie, et al. Tune-a-video: One-shot tuning of image diffusion models for text-to-video generation. In ICCV, 2023.
>
> ---
>
> We would like to gently remind you that the **end of the discussion period is imminent**. We would appreciate it if you could let us know whether our comments addressed your concerns.
>
> Best regards,
> Authors

---

> > ### Comment · Reviewer_NYGw · 2023-11-23
> >
> > Thank you for the author's response. Most of my concerns have been addressed. My remaining reservations primarily pertain to the novelty of the proposed methodology, which resembles an assembled machine, although it has indeed yielded satisfactory results. After careful consideration, I have decided to raise my rating.

---

> > > ### Author Response · Authors · 2023-11-23
> > > **Thank you!**
> > >
> > > Dear Reviewer NYGw,
> > >
> > > Thanks for the positive comments and raising the score.
> > > We are happy to hear that you are satisfied with our answers and that our paper has improved.

---

### Official Review · Reviewer_TXKT · 2023-10-31

**Soundness:** 3 good
**Presentation:** 3 good
**Contribution:** 3 good
**Rating:** 6
**Confidence:** 3

**Summary:**

Ground-A-Video proposes a training-free framework for grounded multi-attribute editing. The grounding ability is achieved by introducing the pretrained GLIGEN gated self-attention module into existing video editing pipelines. The paper also proposes several techniques including "cross-frame attention" and "flow-guided latent smoothing" to improve temporal consistency.

**Strengths:**

1. Grounded video editing is a novel task. Bounding box grounding allows users to accurately select the regions to edit. It is a useful feature that allows better location controllability and content disentanglement over the existing popular sentence-level video editing works.

2. The overall framework is training-free. The proposed framework is built upon several pretrained models like Stable Diffusion, Control Net, GLIGEN, and training-free inversion techniques like Null-text Inversion, so the framework itself requires no additional training on video data.

3. The framework effectively associates editing prompts with the grounded areas. The results in the paper show better text alignment compared to previous sentence-level video editing approaches like Tune-A-Video, and ControlVideo.

**Weaknesses:**

1. Although under the same framework, the technical contributions are quite disconnected from one another and some of them might not be closely related to the grounded video editing task. Two main technical contributions (Modulated Cross-Attention and Flow-guided Latent Smoothing) in the paper lie in finding temporally smooth and faithful latent noise during inversion, which serves as a preparatory step and seems to be less relevant to the grounded editing task. On the other hand, the grounded editing capability mainly comes from the pretrained GLIGEN gated self-attention module, which brings limited addition to the previous image grounding task.

2.  Although the proposed framework is training-free, it is still noteworthy that the per-frame null-text inversion requires gradient-based optimization on the null-text embeddings and could be time-consuming for longer videos. Moreover, the new Modulated Cross-Attention mechanism requires jointly optimizing all frames, which requires large memory.

3. I am not very clear about the flow-guided smoothing after reading the description: does it only work on static areas? If not, why there are not any warping operations mentioned? I also find the terms "spatially continuous" and "spatially discrete" a bit confusing and hard to understand what continuous and discrete refers to.

*For reproducibility, the code has not been released in the provided link at the point of this review.

**Questions:**

Please kindly address the questions in the weakness section.

**Details Of Ethics Concerns:**

n.a.

---

> ### Author Response · Authors · 2023-11-19
> **Reply to Reviewer TXKT (1/2)**
>
> We appreciate the reviewer for the insightful comments.
>
> ---
> >**W1.** Although under the same framework, the technical contributions are quite disconnected from one another and some of them might not be closely related to the grounded video editing task. Two main technical contributions (Modulated Cross-Attention and Flow-guided Latent Smoothing) in the paper lie in finding temporally smooth and faithful latent noise during inversion, which serves as a preparatory step and seems to be less relevant to the grounded editing task. On the other hand, the grounded editing capability mainly comes from the pretrained GLIGEN gated self-attention module, which brings limited addition to the previous image grounding task.
>
> We sophisticatedly designed the technical contributions (Gated Self-Attention into Cross-Frame Gated Attention, Cross-Attention into Modulated Cross Attention and introduction of inflated ControlNet [2] and Optical Flow Smoothing), all connected towards the *common goal of ensuring temporal consistency even in complex editing scenarios*. As the reviewer stated, the grounded editing capability primarily comes from the GLIGEN [1] gated self-attention module, however if directly applied in the Image editing manner, significant inconsistencies are introduced. Please refer to the added **Figure 9**.
>
> We would like to emphasize that our work focuses on multi-attribute ‘*video*’ editing, where even if complex editing is performed successfully on each frame, if the appearance consistency (temporal consistency) is broken between frames, the edited video is considered as failure. We have also included a detailed quantitative analysis in **Table 2**, illustrating the effect of each component in our pipeline on achieving temporal consistency.
>
> ---
> >**W2.** Although the proposed framework is training-free, it is still noteworthy that the per-frame null-text inversion requires gradient-based optimization on the null-text embeddings and could be time-consuming for longer videos. Moreover, the new Modulated Cross-Attention mechanism requires jointly optimizing all frames, which requires large memory.
>
> As the reviewer stated, the per-frame null-text inversion requires gradient-based optimizations. In our work, the null-text inversion consumes 40 seconds.
> Furthermore, we would like to inform you that the denoising stage comprising 50 steps takes 1minute and 50 seconds.
> Although, the gradient-based null-text inversion requires non-trivial time, we believe that accelerating this inversion by parallel batch processing (putting the video frames on the batch dimension) is possible.
> Although we haven’t yet implemented the parallel null-text inversion for videos due to the limitation on computational resource, we will include the inversion on our code release as soon as we implement it.
>
> To further discuss the memory consumption related to the Modulated Cross-Attention, we would like to kindly inform you that the GPU vRAM consumption is not increased from the Modulated Cross-Attention. The main vRAM bottleneck in our framework comes from computation of Cross-Frame Gated Attention, which consumes 20 GB, which is available on a single Quadro RTX 6000 GPU, where we conducted all our experiments.

---

> ### Author Response · Authors · 2023-11-19
> **Reply to Reviewer TXKT (2/2)**
>
> >**W3.** I am not very clear about the flow-guided smoothing after reading the description: does it only work on static areas? If not, why there are not any warping operations mentioned?
>
> The proposed Optical Flow guided Latents Smoothing works on static regions, the regions with relatively small motion difference, which is why we utilized RAFT optical flow estimator [3]. To enhance understanding of the algorithm, we revised a few notations in **Algorithm 1**.
>
> In the previous algorithm, the obtained mask denotes dynamic regions $map_{mask}^{i} \xleftarrow{}  map_{mag}^i>M_{thres}$,
> which performs smoothing on static regions by ${z}^i_T = ({z}^i_T - {z}^{i-1}_T) \* map\_{mask}^i + {z}^{i-1}_T$ .
>
> In the revised algorithm, the obtained mask denotes static regions $map_{mask}^{i} \xleftarrow{}  map_{mag}^{i}<M_{thres}$,
> which performs smoothing on static regions by ${z}^i_T = {z}^{i-1}_T \* map\_{mask}^i + {z}^{i}_T * (1-map\_{mask}^i)$.
>
>
> We would like to inform the reviewer that both the previous algorithm and the revised algorithm perform the exact same operations. The former is designed to be 'code-friendly,' while the latter is optimized for reader-friendliness. Additionally, our framework does not involve any warping operations. We are happy to revise our algorithm to address the reviewer’s concern and we would like to provide more clarifications if needed.
>
> ---
> >**W4.** I also find the terms "spatially continuous" and "spatially discrete" a bit confusing and hard to understand what continuous and discrete refers to.
>
> Spatially-continuous conditions refer to the spatial conditions that explicitly provide structural information by itself, such as edge maps and depth maps. These conditions are strictly spatially aligned with the generated output image. Also, these conditions are stored in a continuous spatial format, such as a 2D image (C x H x W) for depth maps.
> In contrast, Spatially-discrete conditions do not require strict spatial alignment with the output image. In our work, the grounding conditions does not enforce particular structural constrains inside the bounding boxes, but rather determines the positions of entities enclosed by the bounding box within the overall image. While our work visualizes groundings as bounding boxes in each image (e.g., Figure 1), these bounding boxes are stored in a discrete spatial format, represented by the four coordinates that define each bounding box.
> We believe the visualization on Figure 13 will facilitate understanding the explanation above.
> We kindly refer the Reviewer to **Section F.1** Discrete and Continuous Conditions and **Figure 13** in the Appendix.
>  We would be happy to engage in further clarifications if needed.
>
>
> ---
> >For reproducibility, the code has not been released in the provided link at the point of this review.
>
> We regret the delay in releasing the code. We understand the importance of open-source work and we assure that the code will be made publicly available. Also, in the abstract, we have updated the statement “codes are provided at our project page” to “codes will be released.”
>
> ---
> [1] Li, Yuheng, et al. Gligen: Open-set grounded text-to-image generation. In CVPR, 2023.
>
> [2] Zhang, Lvmin, et al. Adding conditional control to text-to-image diffusion models. In ICCV, 2023.
>
> [3] Mokady, Ron, et al. Null-text inversion for editing real images using guided diffusion models. In CVPR, 2023.
>
> [4] Teed, Zachary, and Jia Deng. Raft: Recurrent all-pairs field transforms for optical flow. In ECCV, 2020.

---

> ### Author Response · Authors · 2023-11-21
> **Dear Reviewer TXKT**
>
> As the deadline for the Reviewer-Author discussion phase is fast approaching (there is only a day left), we respectfully ask whether we have addressed your questions and concerns adequately.

---

> > ### Comment · Reviewer_TXKT · 2023-12-04
> > **Thanks for the reply**
> >
> > My concerns regarding the computation cost and the flow-guided smoothing has been addressed. I will keep my rating.

---

### Official Review · Reviewer_vcyz · 2023-10-31

**Soundness:** 4 excellent
**Presentation:** 3 good
**Contribution:** 3 good
**Rating:** 8
**Confidence:** 4

**Summary:**

The paper presents the ﬁrst grounding-driven video editing framework, which is intended to solve the problem of neglected editing, wrong element editing and temporal-inconsistency in context of the complexities of multi-attribute video editing scenarios. Moreover, the proposed method is training-free which overcomes the obstacle of excessive computational cost on video tasks. Spatial-Temporal Attention, Cross-Frame Gated Attention and Modulated Cross Attention are introduced to further enhance consistency, depth map is used as an additional condition to better preserve structure and 3D information and the binary mask calculated through the optical ﬂow map can help maintain the consistency of the background area. Eﬀectiveness has been proven by suﬃcient experiments and convincing qualitative results.

**Strengths:**

S1. This paper presents the ﬁrst training-free grounding-driven video editing framework, which is relatively innovative.

S2. The proposed method and experimental results are consistent with their motivation and eﬀectively solve the problem of multi-attribute video editing in complex scenes.

S3. The method is clearly stated, and the details are comprehensive.

**Weaknesses:**

W1. Since the introduced depth map and optical ﬂow map both reﬂect pixel-level structural information, will this cause the structure before and after editing to be too consistent, so that if the foreground is replaced by an object with inconsistent structure, the editing result will be poor and lack of ﬂexibility? e.g. replace the “rabbit” in the phrase "A rabbit is eating a watermelon on the table" with an animal without long ears.

W2. For similar reasons, this may also limit the editing method to the task of adding or deleting objects.

**Questions:**

See weaknesses above.

**Details Of Ethics Concerns:**

None.

---

> ### Author Response · Authors · 2023-11-19
> **Reply to Reviewer vcyz**
>
> We thank the reviewer for the insightful points.
>
> ---
> >**W1.** Since the introduced depth map and optical ﬂow map both reﬂect pixel-level structural information, will this cause the structure before and after editing to be too consistent, so that if the foreground is replaced by an object with inconsistent structure, the editing result will be poor and lack of ﬂexibility? e.g. replace the “rabbit” in the phrase "A rabbit is eating a watermelon on the table" with an animal without long ears.
>
> Our proposed optical flow smoothing make the static regions in the generated consecutive frames to be similar, whereas shapes of the foreground objects are mainly affected by **ControlNet's guidance** (depth).
> As the reviewer stated, ControlNet [1] injects the pixel-level structural information, where the structural preservation between input image and output is the core advantage, yet weakness of ControlNet.
> Since we employ the ControlNet (Inflated ControlNet), our framework also share the advantage and weakness of the Image ControlNet.
>
> However, in the process of injecting the ControlNet's output features to the main Stable Diffusion [2] UNet's transformer blocks, *we multiply the ControlNet's output features by a scaling term, ‘**ControlNet Scale**’*, before they are added to the main UNet's output features.
> We depict this process in the revised **Figure 3 (Right)** and added the description in **Section 3.3** Inflated ControlNet and **Section 5** Conclusion (Limitation).
> By adjusting this scaling hyperparameter, ranging from 0 to 1, the control over the structural flexibility is gained.
> We also added a visual ablation of it on **Figure 8 and 24**, where we experiment replacing the “rabbit” with an animal without long ears.
>
> ---
> >**W2.** For similar reasons, this may also limit the editing method to the task of adding or deleting objects.
>
> We kindly refer the reviewer to the following examples in our manuscript:
>  - ‘Mountains’ are removed and completely differently-shaped ‘sunset’ or ‘fireworks’ are added in the second video case of **Figure 9**.
>  - Background ‘trees’ successfully removed in the last row of **Figure 21**.
>  - The “lake” is successfully added in the last four rows of **Figure 23**.
>
> We would be happy to engage in further requests or clarifications if needed. Again, we appreciate the reviewer for the sharp observation.
>
> ---
> [1] Zhang, Lvmin, et al. Adding conditional control to text-to-image diffusion models. In ICCV, 2023.
>
> [2] Rombach, Robin, et al. High-resolution image synthesis with latent diffusion models. In CVPR, 2022.

---

> ### Author Response · Authors · 2023-11-21
> **Dear Reviewer vcyz**
>
> As the deadline for the Reviewer-Author discussion phase is fast approaching (there is only a day left), we respectfully ask whether we have addressed your questions and concerns adequately.

---

### Official Review · Reviewer_F2zK · 2023-11-02

**Soundness:** 2 fair
**Presentation:** 2 fair
**Contribution:** 3 good
**Rating:** 6
**Confidence:** 4

**Summary:**

The paper focuses on an interesting problem, namely multi-attribute editing in the video domain. The proposed method relies on recent techniques, such as GLIP, and ControlNet, and integrates the grounding information to perform a sequence of editing operations. The aim is also to propose a training-free pipeline by using pre-trained models in a zero-shot setting. For video-level editing, a stable diffusion backbone is extended for video data with DDIM inversion, and optical flow and depth maps are integrated as conditional to improve video editing quality. The evaluation is conducted on a set of videos and the evaluation is performed qualitatively with a comparison to SOTA and quantitatively by a user study with 28 participants.

**Strengths:**

The topic of paper, multi attribute editing in videos, is a challenging and interesting problem.

The paper seems mostly the integration and extension on available recent T2I pipelines for video domain. But, the selected models are recent and they fit well within the proposed pipeline. Moreover, the pipeline avoids additional training stages that is good for zero-shot pipeline.

**Weaknesses:**

The paper is a well-designed integration of mostly existing techniques for video editing pipeline. The authors explain the steps in subsections with details. However, I found the overall text flow confusing as the stable-diffusion model and the layers of it are explained in multiple  sections, e.g. 3.2 and 3.4, rather than a whole. Additionally, the integrations of the controlNet and optical flow into the whole pipeline are not clear. I think a revised figure with math representations consistent with the text may help to explain the model better.

A more detailed qualitative evaluation of the model could be presented in the experimental section. For instance, to assess the impact of a particular component on the pipeline, the ablation section (Section 4.3) only includes the edited video outputs generated using with and without this component. However, this evaluation could also be conducted quantitatively (as in Table 1) to see the impact of some evaluated components on the pipeline.

**Questions:**

What is the video set used for comparison and evaluation in Table 1?

How similar are the paper's evaluation metrics with CAV (Chen et.al 2023)?

---

> ### Author Response · Authors · 2023-11-19
> **Reply to Reviewer F2zK**
>
> We appreciate the reviewer for very thoughtful and constructive comments.
>
> ---
> >**W1.** I found the overall text flow confusing as the stable-diffusion model and the layers of it are explained in multiple sections, e.g. 3.2 and 3.4, rather than a whole.
>
> We revised the composition of **Section 3** accordingly. The revised composition is as follows:
> - [Section 3.1] Ground-A-Video: Overview
> - [Section 3.2] Inflated Stable Diffusion Backbone
>     - [Subsection 1] Attention Inflation with Spatial-Temporal Self-Attention
>     - [Subsection 2] Per-frame Inversion with Modulated Cross-Attention
>     - [Subsection 3] Video Groundings with Cross-Frame Gated Attention
> - [Section 3.3] Inflated ControlNet
> - [Section 3.4] Optical Flow Guided Inverted Latents Smoothing
>
> ---
> >**W2.**  Additionally, the integrations of the controlNet and optical flow into the whole pipeline are not clear. I think a revised figure with math representations consistent with the text may help to explain the model better.
>
> To clarify the integrations of the pipeline components, we revised a figure with consistent math representations in **Figure 3**. We split the overall pipeline into *Input Preparation* and *Denoising Process* and changed the caption of the figure. We would be happy to engage in further clarifications if needed.
>
> ---
> >**W3.** A more detailed qualitative evaluation of the model could be presented in the experimental section. For instance, to assess the impact of a particular component on the pipeline, the ablation section (Section 4.3) only includes the edited video outputs generated using with and without this component. However, this evaluation could also be conducted quantitatively (as in Table 1) to see the impact of some evaluated components on the pipeline.
>
> To assess the individual impact of each component within the pipeline, we conducted a quantitative analysis presented in **Table 2**. We evaluated text alignment and frame consistency under the condition of omitting each component.
> |                    | Text-Align | Frame-Con |
> |--------------------|------------|-----------|
> | w/o Modulated CA   | 0.835      | 0.967     |
> | w/o Groundings     | 0.802      | 0.960     |
> | w/o Cross-Frame GA | 0.829      | 0.956     |
> | w/o ControlNet     | 0.823      | 0.948     |
> | Full components    | **0.837**  | **0.970** |
>
>
> Moreover, quantitative analysis on Optical Flow Smoothing component is added in the corresponding paragraph in **Section 4.3**.
> | Threshold | 0.2   | 0.3   | 0.4   |
> |-----------|-------|-------|-------|
> | Frame-Con | **0.970** | 0.968 | 0.964 |
>
>
> Coupled with the above suggestions on reorganizing the sections and revising the figure, we believe the reviewer’s feedback added a lot of value to our work and sincerely appreciate the time devoted to deeply analyzing our manuscript.
>
> ---
> >**Q1.** What is the video set used for comparison and evaluation in Table 1?
>
> We used a subset of 20 videos from DAVIS [1] dataset. We kindly refer the reviewer to **Section 4.1** Implementation Details. We would like to further clarify the video clips used in our experiments if needed.
>
> ---
> >**Q2.** How similar are the paper's evaluation metrics with CAV (Chen et.al 2023)?
>
> CAV [2] employed XCLIP [3] video encoder to obtain video embeddings then computed cosine similarity between the targe text embeddings. In contrast, our work employed CLIP [4] image encoder to obtain embeddings of each frame, computed cosine similarities between the target text embeddings, then averaged the cosine similarities.
>
> ---
> [1] Pont-Tuset, Jordi, et al. The 2017 davis challenge on video object segmentation. In arXiv preprint, 2017.
>
> [2] Chen, Weifeng, et al. Control-A-Video: Controllable Text-to-Video Generation with Diffusion Models. In arXiv preprint, 2023.
>
> [3] Ni, Bolin, et al. Expanding language-image pretrained models for general video recognition. In ECCV, 2022.
>
> [4] Radford, Alec, et al. Learning transferable visual models from natural language supervision. In ICML, 2021.

---

> ### Author Response · Authors · 2023-11-21
> **Dear Reviewer F2zK**
>
> As the deadline for the Reviewer-Author discussion phase is fast approaching (there is only a day left), we respectfully ask whether we have addressed your questions and concerns adequately.

---

> ### Author Response · Authors · 2023-11-23
> **[Reminder] Summarization of our rebuttal**
>
> Dear reviewer F2zK,
>
> We believe that we have addressed the concerns that you have raised. Specifically,
>
> 1. We have restructured **Section 3** (Method) in response to the your feedback regarding the text flow.
> Specifically, we have consolidated the explanation of the inflated Stable Diffusion model [1] and its layers into a cohesive presentation in Section 3.2. Additionally, we have provided elucidation of the Inflated ControlNet [2] in Section 3.3 and introduction  of Optical Flow-guided Smoothing in Section 3.4, respectively.
>
> 2. In response to the your suggestions, we have made revisions to **Figure 3** (Overview of Ground-A-Video pipeline) to ensure consistent mathematical representation and enhance the clarity of the pipeline component integrations. More specifically, we have subdivided the entire pipeline into the *Input Preparation Process* and *Denoising Process*, and we have adjusted the figure's caption to reflect these changes.
>
> 3. We have expanded our qualitative evaluation of Ground-A-Video in **Table 2**.
> In order to assess the effects of various components, including Modulated Cross Attention (vs Cross Attention), Video Groundings guidance (vs Absence of Groundings guidance), Cross-Frame Gated Attention (vs Gated Self-Attention), and Inflated ControlNet guidance (vs Absence of ControlNet guidance), on our pipeline, we conducted evaluations of text alignment and frame consistency under each of these conditions.
> Furthermore, we have introduced quantitative analysis of our proposed Optical Flow-guided Inverted Latents Smoothing in the relevant section of Section 4.3.
>
> 4. We have clarified the video dataset used for the comparison and evaluation. Additionally, we have provided an explanation of the similarities and differences between our Frame Consistency metric and the Frame Consistency metric used by CAV [3].
>
>
> [1] Rombach, Robin, et al. High-resolution image synthesis with latent diffusion models. In CVPR, 2022.
>
> [2] Zhang, Lvmin, et al. Adding conditional control to text-to-image diffusion models. In ICCV, 2023.
>
> [3] Chen, Weifeng, et al. Control-A-Video: Controllable Text-to-Video Generation with Diffusion Models. In arXiv preprint, 2023.
>
> ---
>
> We would like to gently remind you that the **end of the discussion period is imminent**.
> We would appreciate it if you could let us know whether our comments addressed your concerns.
>
> Best regards,
> Authors

---

> > ### Comment · Reviewer_F2zK · 2023-11-23
> >
> > Thanks for your detailed rebuttal which addresses many points I raised. But, I still have one major concern that the current format of the submission may require substantial revision particularly in the text part. I will evaluate your rebuttal again and your answers to other reviewer's comments.

---

> > > ### Author Response · Authors · 2023-11-23
> > > **Reply to the Comment by Reviewer F2zK**
> > >
> > > Dear Reviewer F2zK
> > >
> > > Thank you for your feedback.
> > >
> > >  In response to your previous suggestion to reorganize the Method Section, we have already revised Section 3. The restructuring involved rearranging the order of the subsections without altering their individual content.
> > >
> > > We appreciate your feedback but are seeking some clarification regarding your comment on the "current format of the submission may require substantial revision particularly in the text part."
> > >
> > > Could you please specify which version of our manuscript your comment refers to?
> > > Are you referring to the revised version or the initial submission when mentioning 'the current format'?
> > >
> > > If you are referring to the revised version of our manuscript, we are more than willing to implement further modifications in the text sections as specified by you.
> > >
> > > Best,
> > > Authors

---

> > > > ### Comment · Reviewer_F2zK · 2023-11-23
> > > >
> > > > Thanks for asking the clarification. I meant by "substantial" the revision over the "initial" submission. I will carefully check your revised version before finalizing my decision.

---

> > > > > ### Author Response · Authors · 2023-11-23
> > > > > **Thanks for the clarification**
> > > > >
> > > > > Dear Reviewer F2zK,
> > > > >
> > > > > Thanks for the clarification.
> > > > >
> > > > > We would like to assure the reviewer that although we have made changes of the text, it is mainly for the structural change (reorganizing the hierarchy of subsection, sections) for clarification rather than the extensive amount of content changes.
> > > > > We would like to also remind that the other reviewer has just raised the score by commenting that the revision was satisfactory.
> > > > >
> > > > > Hope this can address your concern.
> > > > >
> > > > > Best,
> > > > >
> > > > > Authors

---

### Author Response · Authors · 2023-11-19
**General Response (1/2)**

We thank all the reviewers for their valuable reviews.
We also appreciate their recognition of the key contributions of our framework and the superiority of our results.

1. **Contributions** to the task of multi-attribute video editing:
    - “The paper focuses on an interesting problem, namely multi-attribute editing in the video domain.”, “The topic of paper, multi attribute editing in videos, is a challenging and interesting problem.” (Reviewer F2zK)
    - “This paper addresses a highly significant problem of achieving consistent fine-grained attribute editing in videos.” (Reviewer NYGw)

2. The **novelty** of Ground-A-Video:
    - “The paper presents the ﬁrst grounding-driven video editing framework.” (Reviewer vcyz)
    - “Grounded video editing is a novel task.” (Reviewer TXKT)
    - “In addition to the design of grounding conditions, the authors propose mechanisms such as noise smoothing and cross-attention.” (Reviewer NYGw)

3. **Superior performance** of Ground-A-Video:
    - “The proposed method and experimental results are consistent with their motivation and eﬀectively solve the problem of multi-attribute video editing in complex scenes.” (Reviewer vcyz)
    - “The results in the paper show better text alignment compared to previous sentence-level video editing.” (Reviewer TXKT)
    - “The introduction of grounding conditions proves to be a direct and effective approach.” (Reviewer NYGw)

4. The merit of **training-free** framework:
    - “The pipeline avoids additional training stages that is good for zero-shot pipeline.” (Reviewer F2zK)
    - “This paper presents the ﬁrst training-free grounding-driven video editing framework, which is relatively innovative.” (Reviewer vcyz)
    - “The framework itself requires no additional training on video data.” (Reviewer TXKT)

---

> ### Author Response · Authors · 2023-11-19
> **General Response (2/2)**
>
> Here, we have summarized changes to our manuscript. Modified or added contents have been highlighted blue in the revised paper. Point-to-point responses were also included as a reply to each reviewer.
>
> ---
>
> ### 1. Figure 3:
> We revised the pipeline overview figure (**Figure 3**) and the corresponding caption.
> In specific, we separated *Input Preparation* and *Denoising Process* for better readability.
>
>
> ### 2. Section 3:
> We reorganized the Method section (**Section 3**).
> Accordingly, we modified the paragraph in **Section 3.1** where we briefly explain the order of upcoming contents.
>
>
> ### 3. Algorithm 1:
> We modified notations of **Algorithm 1** (Optical Flow guided Inverted Latents Smoothing).
> Previous lines of  $map_{mask}^{i} \xleftarrow{}  map_{mag}^i>M_{thres}$ and ${z}^i_T = ({z}\_T^i - {z}\_T^{i-1}) \* map\_{mask}^i + {z}\_T^{i-1}$
> are revised into
> $map_{mask}^{i} \xleftarrow{}  map_{mag}^{i}<M_{thres}$
> and
> ${z}^i_T = {z}^{i-1}_T \* map\_{mask}^i + {z}^{i}_T * (1-map\_{mask}^i)$.
> It's worth noting that both the previous algorithm and the revised algorithm perform the exact same operations. The previous algorithm closely mirrors the code-level implementation, while the revised algorithm aligns more effectively with the description in the paragraph (**Section 3.4**) that outlines the algorithm. This alignment is particularly notable because the obtained binary mask in the revised algorithm represents the static region.
>
>
> ### 4. Figure 8, 26 and Section 3.3:
> We added a result ablating flexibility of structure preservation between input and output in **Figure 8 and 26** and added discussion on how to achieve the flexibility in **Section 3.3**.
>
>
> ### 5. Table 2:
> We added quantitative experiments and results on each pipeline component in **Table 2**.
>
>
> ### 6. Section 4.3 and Figure 6:
> We added a quantitative analysis on **Optical Flow Smoothing Subsection in Section 4.3** and added additional examples of the proposed flow smoothing with varying thresholds in **Figure 6 (Left)**.
>
>
> ### 7. Figure 9:
> We added comparisons with per-frame Stable Diffusion [2] based editing methods, particularly GLIGEN [3] and ControlNet[4] in **Figure 9**.
>
>
> ### 8. Figure 10:
> We added a result of applying the Optical Flow Smoothing in Tune-A-Video [1] to demonstrate robustness of the proposed method in **Figure 10**.
>
> ---
>
> [1] Wu, Jay Zhangjie, et al. Tune-a-video: One-shot tuning of image diffusion models for text-to-video generation. In ICCV, 2023.
>
> [2] Rombach, Robin, et al. High-resolution image synthesis with latent diffusion models. In CVPR, 2022.
>
> [3] Li, Yuheng, et al. Gligen: Open-set grounded text-to-image generation. In CVPR, 2023.
>
> [4] Zhang, Lvmin, et al. Adding conditional control to text-to-image diffusion models. In ICCV, 2023.

---

### Meta-Review · Area_Chair_CuSn · 2023-12-05

**Metareview:**

The paper receives mixed reviews in the beginning. Most reviewers appreciate the innovative task design and effective results. However, there are some concerns about the limited novelty as the work is mainly recomposing existing techniques.
During the rebuttal, the authors addressed most of the concerns, including adding more results and improve the text writing. All reviewers recommend acceptance after the discussion. The authors are encouraged to address any remaining concerns for camera-ready.

**Justification For Why Not Higher Score:**

Although the task is interesting and results are satisfactory, the individual components are not very innovative.

**Justification For Why Not Lower Score:**

The authors propose an interesting task, as acknowledged by all reviewers, and a simple yet effective way to solve the problem.

---

### Decision · Program_Chairs · 2024-01-16

Accept (poster)